# Effect of the Foundation Modelling on the Fatigue Lifetime of a Monopile-based Offshore Wind Turbine

Steffen Aasen[1], Ana M. Page[2,3], Kristoffer Skjolden Skau[2,3], Tor Anders Nygaard[1]

[1]Institute for Energy Technology, PO Box 40, 2027 Kjeller, Norway
[2]Norwegian Geotechnical Institute, Sognsveien 72, 0855 Oslo, Norway
[3]Norwegian University of Science and Technology, Department of Civil and Transport Engineering, Høgskoleringen 7A, 7491 Trondheim, Norway

*Correspondence to*: tor.anders.nygaard@ife.no

**Abstract.** Several studies have emphasized the importance of modelling foundation response with representative damping and stiffness characteristics in integrated analyses of offshore wind turbines (OWT's). For the monopile foundation, the industry standard for pile-analysis has shown to be inaccurate, and alternative models that simulate foundation behaviour more accurately are needed. As fatigue damage is a critical factor in the design phase, this study investigates how four
different soil-foundation models affect the fatigue damage of an OWT with monopile foundation. The study shows how both stiffness and damping properties have a noticeable effect on the fatigue damage, in particular for idling cases. At mudline, accumulated fatigue damage varied up to 22% depending on the foundation model used.

# 1   Introduction

The last decade there has been a strong tendency to look offshore to further increase the wind energy potential in Northern Europe. This had led to a total of over 3000 installed offshore wind turbines, with a capacity of more than 11GW (December 2015). Large offshore sites with suitable wind conditions are still accessible, and together with strong political incentives this lead to high growth expectations for the industry (EWEA, 2015a). Installation, maintenance and foundation costs tend to increase with distance to shore and water depth, making cost reductions important. So far, improved supply chain integration and large capacity turbines have been the main methods for cost reductions (ORE Catapult, 2015). However, cost reductions can also be achieved by cost-efficient design. With the support structure contributing up to 20% of the capital cost (EWEA, 2015b), optimizing foundation design has a high potential for cost reductions.

Integrated time domain analysis plays a central role in the design phase of OWT's. Integrated analysis refers to fully coupled analysis of the complete OWT system, including rotor, support structure and foundation. The foundation response has significant impact on the dynamic behaviour of the OWT (see Sect. 2.3), which as a result influence the dimensioning of the structure. As a consequence, the foundation modelling becomes important in the design phase, as integrated time domain analysis are used.

For depths up to 30 meters, the monopile is the most common support structure, accounting for approximately 80% of the installations (EWEA, 2015a). This foundation type result in long and slender structures sensitive to resonance effects, since wave and wind loads are typically close to the natural frequencies of the structure. Because of this, the soil-foundation response can have a high impact on the dynamics of the system and thereby the fatigue damage of the structure. With fatigue damage being a design driver, soil-foundation modelling becomes important in design. The most widespread methods for fatigue estimations are time domain simulations with S-N curves, and frequency domain calculations. A comparison of these methods can be found in Ragan and Manuel (2007). The industry standard for fatigue damage calculations is the time domain simulations with S-N curves described by DNV (Det Norske Veritas, 2014), which is used in this study.

Different approaches can be used to model the soil-foundation response for piles. Generally, they are divided into two groups: continuum approaches and subgrade reaction approaches. In continuum approaches, the soil is treated as a continuum material described by a constitutive relation. The problem of a pile embedded in a continuum material can be solved analytically if the soil is assumed to be a linear-elastic material, e.g. Poulos (1971), or numerically if the soil is characterized by a more complex constitutive relation. Among the numerical methods, the boundary element method, see for instance Kaynia and Kausel (1982), and the finite element method, e.g. Randolph (1981) or Andresen et al. (2010), are the most widely used. In the subgrade reaction approaches, the soil response around the pile is described by a set of uncoupled individual horizontal springs, where the interaction between layers is only taken into account by the pile continuity. The springs relate the local lateral resistance, $p$, to the local lateral displacement of the pile, $y$, following a predefined function.

Several *p-y* functions can be found in the literature, see for instance Reese and Van Impe (2010), however, the API (2011) *p-y* curves are the most widely used.

The aim of this paper is to study four different soil-foundation models with respect to their impact on fatigue damage of the OWT structure. Both the conventional method for pile-analysis (p-y curves), and simple linear elastic models, have been compared with a non-linear elastic model with hysteretic damping. A range of environmental conditions have been simulated to study the soil-foundation models for different loading. The paper is a continuation of the master's thesis of the first author (Aasen, 2016).

Following the introduction, Chapter 2 gives a review of observed foundation behaviour, current foundation models for OWT monopiles, and relevant studies investigating effects of soil stiffness and damping. Chapter 3 presents the simulation software 3DFloat and the different soil-foundation models studied in this paper. Chapter 4 presents the OWT structure, the soil profile and the environmental conditions that have been applied in simulations. The calibration of each soil-foundation model, together with the methodology for fatigue damage calculations are also included at the end of Chapter 4. Chapter 5 presents the results from the analysis that has been carried out, followed by the conclusion in Chapter 6.

## 2 Foundation behaviour

### 2.1 Observed foundation behaviour

The foundation has to resist the loads transferred from the structure above and remain functional and stable during the lifetime of the OWT. Piles supporting monopile-based OWTs are subjected to large horizontal loads applied with an arm of about 30 – 90 m, which results in large bending moments at the foundation. The vertical load is relatively small compared with the horizontal and bending moment loads (Byrne and Houlsby, 2003). Large diameter piles resist these loads by mobilizing lateral resistance in the soil. Due to the interaction between the pile and the soil, the foundation response is influenced by the response of the soil around it. The most important characteristics of soil behaviour with respect to monopiles are:

(1) **Non-linear response**. Soils show non-linear response during loading. In pile foundations, the generation of plastic deformations in the soil around the pile causes plastic displacements and rotations, resulting in a non-linear load-displacement foundation response. This behaviour is illustrated in Figure 1 between points 0 and 1. Several pile tests displaying the non-linear load-displacement response can be found in the literature, see for instance Poulos and Davis (1980), Cox et al. (1974), Reese et al. (1975) for flexible piles or Byrne et al. (2015) for more rigid piles with large diameters typical for monopiles supporting monopile-based OWTs.

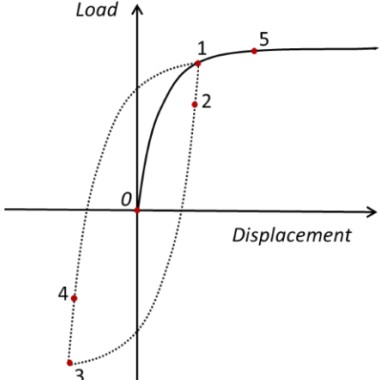

**Figure 1: Observed foundation behaviour.**

(2) **Different stiffness during loading, unloading and reloading**. Soils exhibit different stiffness during loading, unloading and reloading. When the load acting on the foundation is reversed (points 1 to 2 in Figure 1), the soil around the pile is unloaded. Initially the soil unloading is elastic and the pile response is stiffer than prior to the reversal. As the magnitude of the load reversal increases, more plastic deformations are generated and the stiffness decreases (points 2 to 3). During reloading (points 3 to 5 and back to 1), a similar pattern is observed. This behaviour has been reported in cyclic large- and

small-scale pile tests, see for instance Little and Briaud (1988), Roesen et al. (2013) or in centrifuge tests, e.g. Klinkvort et al. (2010), Bienen et al. (2011) or Kirkwood (2015).

(3) **Damping**. Two different types of damping are present in foundation problems: radiation damping, where the energy is dissipated through geometric spreading of the waves propagating through the soil, and hysteretic damping, where energy is
dissipated due to plastic deformations. Radiation damping depends on the loading frequency, and it is negligible for frequencies below 1 Hz (Andersen, 2010). Hysteretic soil damping depends on the strain level in the soil and is affected by the loading history. For monopiles supporting OWT's, radiation damping can be neglected, and the main damping contribution comes from hysteretic damping. The hysteretic nature of the foundation damping has been noted in free vibration tests (Hanssen et al., 2016), where the foundation damping decreased with decreasing displacement amplitude. The
hysteretic loss of energy at foundation level is illustrated in Figure 1 in the enclosed area defined by points 1-2-3-4-1. Hysteretic load displacement loops can also be observed in cyclic large- and small-scale pile tests and in centrifuge tests (Klinkvort et al., 2010, Roesen et al., 2013).

Full-scale measurements of monopile-based OWTs also confirm the non-linear hysteretic foundation response. Kallehave et al. (2015) observed that the measured natural frequency of monopile-based OWTs decreased with increasing wind speeds,
and related it to increasing displacement levels. The same conclusion was reached by Damgaard et al. (2013) when analysing the reduction in natural frequency with increasing acceleration levels.

For OWT's in operation several authors (Damgaard et al., 2013, Shirzadeh et al., 2013, Tarp-Johansen et al., 2009, Versteijlen et al., 2011) found foundation damping between 0.25-1.5 % of critical damping depending on load level and soil profile.

## 2.2    Current foundation modelling

The industry standard for representing the pile response in integrated analyses of monopile-based OWTs is based in the so-called *p-y* curve approach. In the *p-y* curve methodology, the pile is modelled as a beam and the soil is represented by a series of discrete, uncoupled, non-linear elastic springs at nodal points along the pile. The springs relate the local lateral resistance, *p*, to the local lateral displacement of the pile, *y*, and are function of the depth below mudline. The DNV standard (Det Norske Veritas, 2014) recommends the use of API *p-y* curves (API, 2011) for the estimation of the lateral pile capacity in
ULS analyses. However, the *p-y* curves were developed for long and slender jacket piles with large length-to-diameter ratios, significantly different from typical monopile geometries. Several studies have shown the limitations of the *p-y* curve approach (Doherty and Gavin, 2011, Lesny, 2010, Jeanjean, 2009, Hearn and Edgers, 2010), and alternatives to the API formulation have been proposed, such as *p-y* curves extracted from FE analysis of the soil-foundation system. Despite these curves being able to capture the pile stiffness more accurately, the same extracted *p-y* curve is often used in the simulation
tools for loading, unloading and reloading, which means that they neglect effects such as permanent deformations and soil damping. In this regard, Det Norske Veritas (2014) requires soil-damping to be considered in the design phase, but no recommended practice for estimating soil damping is suggested.

## 2.3    Numerical studies investigating effects of soil stiffness and damping

Some studies have been carried out to investigate the impact of soil stiffness and damping on the structural response of monopile-based OWTs. Schafhirt et al. (2016) examined the effect of variations in the soil stiffness on the equivalent damage loads for a monopile in sand by using *p-y* curves with different stiffness. The study suggest that a reduction of 50% in the soil stiffness lead to an increase of 7% in the equivalent damage loads at mudline. Damgaard et al. (2015) studied the impact of a change in soil stiffness and damping on the fatigue loads, where the foundation was represented by a lumped-parameter model. They found that a 50% reduction of the soil's Young modulus increased the fatigue damage equivalent moment at mudline by approximately 12%; and a 50% reduction of the soil damping properties increased the fatigue damage equivalent moment by 25%. Carswell et al. (2015) studied the effect of soil damping for an OWT with monopile foundation subjected to extreme storm loading. The hysteretic damping was computed using a non-linear elastic two dimensional finite element model, and included in the foundation model by a viscous rotational damper at mudline. From stochastic time history analysis they found that maximum and standard deviation of mudline moment was reduced by 7-9% due to soil damping. These contributions highlight the impact of the soil stiffness and damping on the fatigue loads. However, each of these studies uses different soil profiles and modelling approaches to represent the foundation stiffness and damping, which makes a comparison between the different foundation models and damping contributions difficult. On this regard, Zaaijer (2006) compared the first and second predicted natural frequencies for different foundation models: an effective fixity length, a linear elastic stiffness matrix at mudline, uncoupled springs and *p-y* curves. The elastic stiffness and the *p-y* curves gave comparable predicted natural frequencies. In addition, Jung et al. (2015) carried out a comparison between three foundation models with focus on the foundation stiffness. In the study, the foundation response was represented by a stiffness matrix at mudline, distributed *p-y* elements and a finite element (FE) model of the soil volume. Foundation damping was neglected. It was found that the bending moments calculated by using the *p-y* approach and the FE approach were very similar.

In this study, the aim is to evaluate the impact of foundation stiffness and damping through the foundation modelling approach on the fatigue damage of a monopile-based OWT in a lifetime perspective. For that purpose, four different foundation models have been calibrated for the same soil profile, and a series of simulations representative for the OWT lifetime have been performed. The results in terms of accumulated fatigue damage are presented and discussed.

# 3 Modelling of OWT and foundation

## 3.1 3DFloat

The simulation software 3DFloat has been used for modal analysis and time domain simulations. 3DFloat is an aero-servo-hydro-elastic Finite-Element-Method code, developed by IFE and NMBU. This means that hydrodynamic loads, aerodynamic loads and the control system are considered when calculating the elastic response of the system. 3DFloat has been verified and validated in the IEA OC3, OC4 and OC5 projects, wave tank tests and by participation in commercial projects. For more details, see Nygaard et al. (2016).

Structural elements are modelled by Euler-Bernoulli beams with 12 degrees of freedom. Loads from gravity, buoyancy, waves, current and wind are applied as distributed external loads on the structure. Forces per unit length are integrated with the interpolation functions used in the Galerkin formulation of the Finite-Element-Method. The distributed forces are thereby lumped to consistent nodal loads (forces and moments), applied to the nodes connecting the elements. Forces from waves and currents on slender beams, are calculated by the relative form of Morison's equation. In this study, combinations of Airy wave components according to the JONSWAP spectrum were used to simulate irregular sea states.

The quadratic drag forces on the tower above the instantaneous wave surface are computed from the turbulent wind. The wind turbine distributed blade loads are computed from Blade Element Momentum theory, taking into account the elastic deformation of the structure.

Currently new soil-foundation models are implemented in 3DFloat as part of the research project REDWIN[1]. Previously springs and dampers were used to model soil resistance. As part of this study, a non-linear model with hysteretic damping has been implemented in the code, referred to as Model 4 throughout this paper.

## 3.2 Soil-foundation models

Four approaches have been used to model the pile-foundation response. They are referred as Model 1, Model 2, Model 3 and Model 4. Model 1 refers to the conventional distributed p-y element approach. Model 2-4 give the full loading response from the soil-pile system at a single node connected to the superstructure. Of the four models, Model 3 and Model 4 account for soil damping. Below follows a description of the four models. The calibration of each model is presented in Section 4.4.

### 3.2.1 Model 1

Model 1 refers to the p-y-element approach where non-linear elastic springs are distributed along the pile and provide soil resistance at different depths. The model does not include any damping. A representation of the model is given in Figure 2. Each spring can have different soil-reaction displacement characteristics. The bottom of the pile is constrained for

---

[1] www.ngi.no/eng/Projects/REDWIN

translation along, and rotation around the z-axis. P-y curves are applied along both horizontal directions. A typical shape of a p-y curve for sand is given in Figure 3.

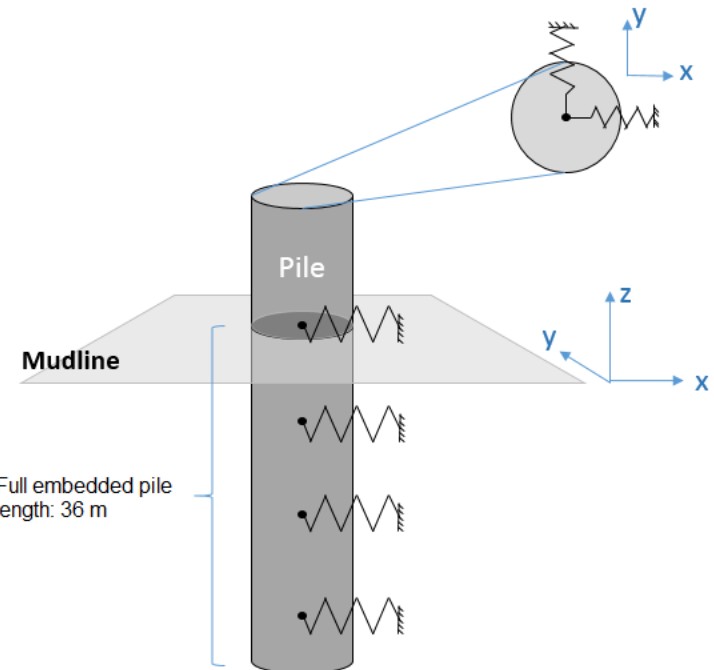

**Figure 2: Model 1 applies non-linear p-y curves along the immersed part of the pile. The p-y curves are applied to both horizontal directions. The bottom of the pile is restrained against translation in the z-direction and rotation around the z-axis.**

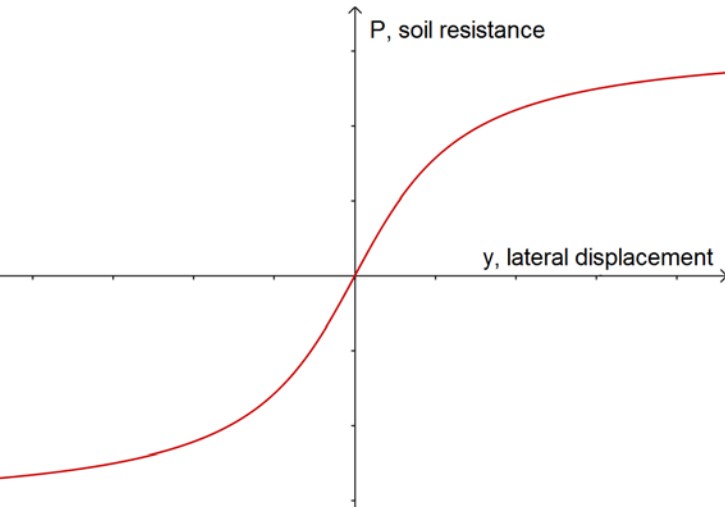

**Figure 3: Typical shape of p-y curves for sand.**

### 3.2.2    Model 2

In Model 2, a linear elastic stiffness matrix applied at the mudline represents the pile-foundation response. By this approach, the model neglects non-linear effects and damping. A representation of the model is given in Figure 4. The stiffness coefficients should reflect the load level considered to best represent modal properties of the system. The stiffness matrix used in this study is presented in Section 4.4.3.

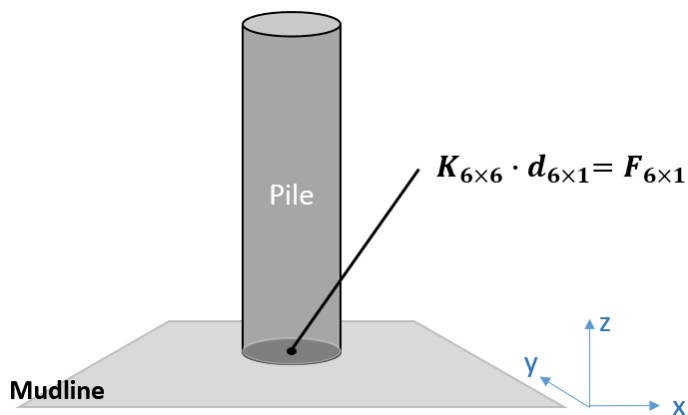

**Figure 4: Model 2 applies a stiffness matrix at the mudline node, to model the soil-foundation response.**

### 3.2.3    Model 3

Model 3 applies a stiffness matrix as in Model 2, but a damping matrix is included to account for soil damping. A representation of this model is given in Figure 5, where $\mathbf{K}$ is a stiffness matrix, $\mathbf{d}$ is a displacement vector and $\mathbf{C}$ is a damping matrix. In Model 3, damping coefficients are only applied in rotational degrees of freedom (rotation around x- and y-axis), as moment typically dominate mudline loading for OWT monopiles (Carswell et al., 2015). The moment response for a single degree of freedom is given by:

$$M_c = c \cdot \dot{\theta}$$
(1)

, where $c$ *[Nm s/rad]* is a damping coefficient, and $\dot{\theta}$ *[rad/s]* is the angular rate of change. This makes soil damping a function of frequency for the system. As explained in Sect. 2.1, hysteretic damping dominates the damping from the foundation. As hysteretic damping is a function of load level and not frequency, the damping coefficients should be calibrated for a given load level and load frequency. Knowing the hysteretic energy loss due to soil damping per load cycle, a damping coefficient for a single degree of freedom can be found by:

$$c = \frac{E_h(M)}{2\theta^2 \pi^2 f}$$
(2)

, where $E_h(M)$[J] is the hysteretic energy loss per load cycle, $\theta$[rad] is the angular displacement amplitude, and $f$[s$^{-1}$] is the loading frequency of the system.

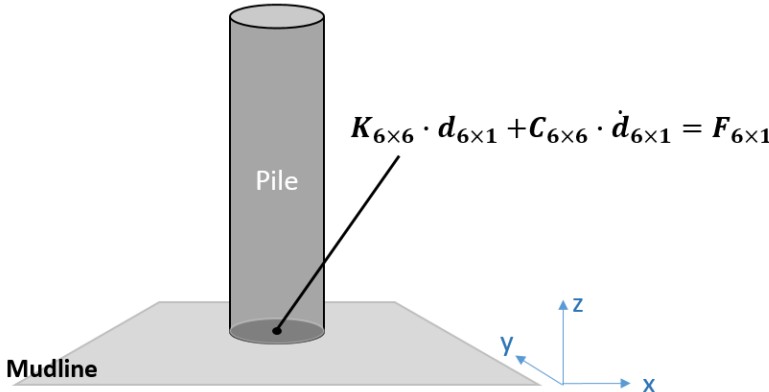

$$K_{6\times6} \cdot d_{6\times1} + C_{6\times6} \cdot \dot{d}_{6\times1} = F_{6\times1}$$

Figure 5: Model 3 applies a stiffness matrix and a damping matrix at the mudline node, to model the soil-foundation response.

### 3.2.4    Model 4

Model 4 is a non-linear 1D rotational model, where the stiffness depends on the load level. The model is applied to rotational DOF's (rotation around x- and y-axis). To reflect both the rotation and horizontal displacement at mudline, the model is applied 10 meters below the mudline, where the pile can rotate around the x- and y-axis. A massless rigid beam connects the model to the flexible tower at mudline. A representation of the model is given in Figure 6.

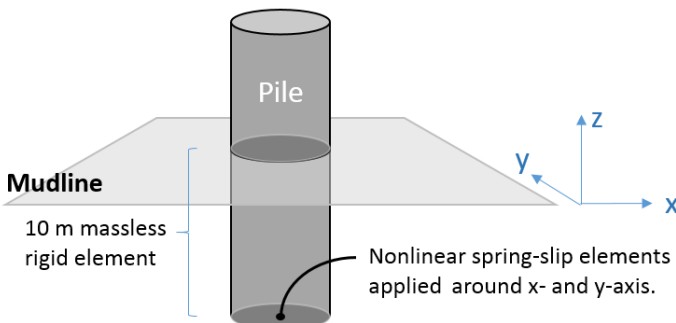

Figure 6: Model 4 applies a 10 m massless rigid element and spring slip elements at the node 10 m below the mudline, to model the soil-foundation response.

The model can reproduce different stiffness during loading, unloading and reloading and produce hysteretic damping. The model is formulated following the approach suggested by Iwan (1967), where several linear elastic-perfectly plastic springs are coupled in parallel, as illustrated in Figure 5a. Each of the springs has different stiffness $k_i$ and yield load $M^*_i$, but forced to have the same deformation $\theta$. The linear elastic-perfectly plastic behaviour of each individual spring and slip element is shown in Figure 5b. The load $M'_i$ of each spring and slip element increases linearly with a stiffness $k_i$ until the critical slipping moment $M^*_i$ is reached. When the loading direction is reversed, each spring and slider element unloads or reloads

following the spring stiffness $k_i$, reproducing Masing's rule (Masing, 1926). The resulting load $M$ is calculated as the sum of each $M_i$. The model gives a stepwise variable stiffness and an overall kinematic hardening behaviour when subjected to cyclic loading. An example of the $M\text{-}\theta$ response is shown in Figure 5c. The load-displacement curve can be represented sufficiently smooth by using a high number of coupled springs. A similar model has been recently implemented in the simulation software FAST (Jonkman and Buhl Jr, 2005) by Krathe and Kaynia (2017).

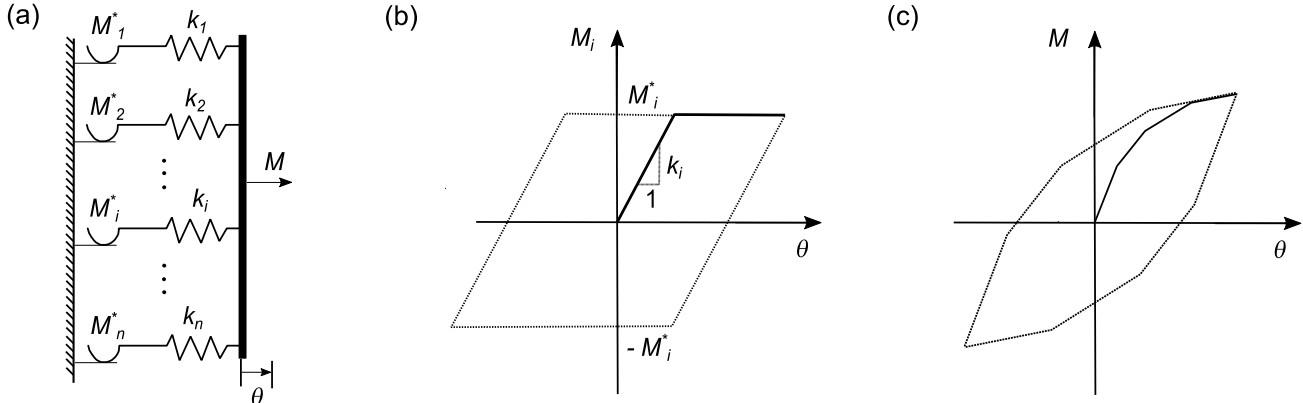

**Figure 7: Non-linear 1D model as a combination of spring-slip elements: (a) physical representation; (b) load-displacement behaviour of each spring and slip element; (c) resulting load-displacement behaviour of all the parallel coupled springs and slip elements.**

# 4    Case Study

## 4.1    Structural properties of OWT

The NREL 5MW Wind turbine, with monopile foundation according to OC3 Phase II (Jonkman and Musial, 2010), has been used in this study. The structure was developed to support concept studies for offshore wind turbines, and is a utility-scale multi-megawatt wind turbine, with a three bladed upwind variable-speed variable-blade-pitch-to-feather-controlled turbine (Jonkman et al., 2009). An overview of the structural dimensions is given in Figure 8. For details about the structure, the reader is referred to (Jonkman et al., 2009).  The transition piece has not been modelled and the pile properties are extended up to the tower in the 3DFloat model.

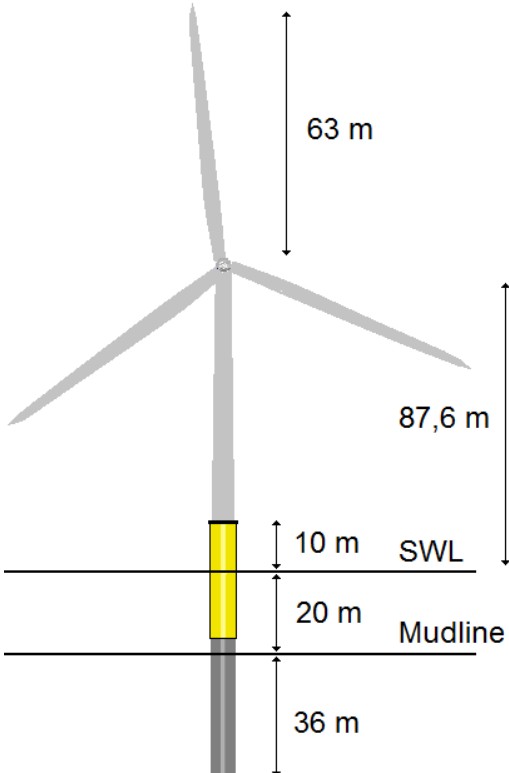

**Figure 8: Dimensions of the NREL 5MW wind turbine, with monopile foundation.**

## 4.2    Soil profile

The soil profile has been taken from OC3, Phase II (Jonkman and Musial, 2010). It is a three layered profile, with effective unit weigh,$\gamma'$,of 10 $[kN/m^3]$ and varying angle of friction, $\phi'$ [°], representing a medium dense sand. The stiffness of the p-y

curves increases proportionally with depth according to the effective stress increase from the weight of soil. A representation of the soil layers and pile dimensions is given in Figure 9.

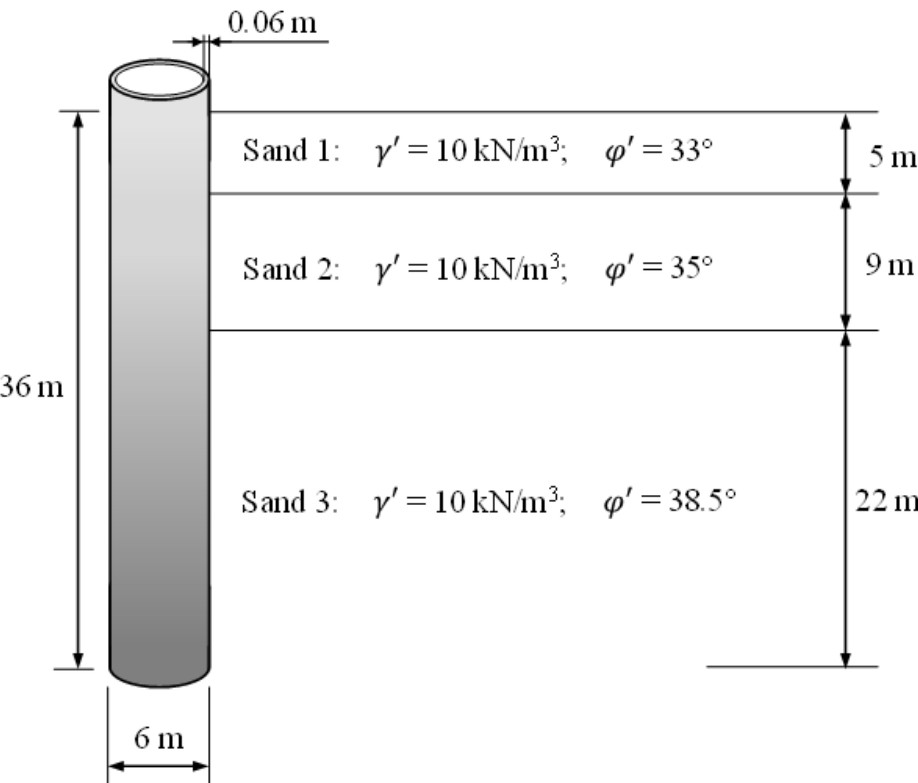

**Figure 9: Soil profile and pile dimensions.**

### 4.3    Environmental conditions

Environmental conditions, representing a possible site for monopile installation in the North Sea have been used in the analyses. A lumped scatter diagram of wind and waves, generated for fatigue damage calculations, was taken from the Upwind Design Basis for a shallow water site with 25m depth (Fischer et al., 2010). The lumped scatter diagram is generated to limit the number of load cases, while giving equivalent fatigue damage to real site wind and wave conditions. Waves and wind are unidirectional, normal to the rotor plane, as the focus of this study has been dynamics in the fore-aft plane. A presentation of wind and wave data is given in Table 1. Turbulence has been generated according to the Mann turbulence model (Mann, 1998). Wind speed is given at the hub height, and a power law, with wind shear exponent of 0.14, gives the wind profile. Superposition of Airy wave components given by the JONSWAP spectrum with a gamma factor of 2.87 is used to generate the irregular wave kinematics.

Figure 10 presents the overturning moment at mudline from wind (+inertial forces) and waves for load case 6. This is near rated conditions for the NREL 5MW wind turbine.

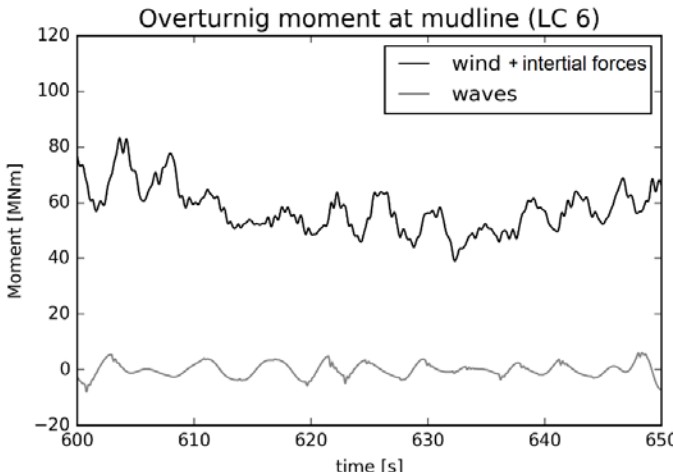

**Figure 10: Overturning moment at mudline from wind and waves from load case 6.**

5 **Table 1: Environmental conditions.**

| Load case | U_wind [m/s] | Ti [%] | Hs [m] | Tp [s] | Pro_occ |
|---|---|---|---|---|---|
| 1 | 2 | 29.2 | 1.07 | 6.03 | 0.06071 |
| 2 | 4 | 20.4 | 1.10 | 5.88 | 0.08911 |
| 3 | 6 | 17.5 | 1.18 | 5.76 | 0.14048 |
| 4 | 8 | 16.0 | 1.31 | 5.67 | 0.13923 |
| 5 | 10 | 15.2 | 1.48 | 5.74 | 0.14654 |
| 6 | 12 | 14.6 | 1.70 | 5.88 | 0.14272 |
| 7 | 14 | 14.2 | 1.91 | 6.07 | 0.08381 |
| 8 | 16 | 13.9 | 2.19 | 6.37 | 0.08316 |
| 9 | 18 | 13.6 | 2.47 | 6.71 | 0.04186 |
| 10 | 20 | 13.4 | 2.76 | 6.99 | 0.03480 |
| 11 | 22 | 13.3 | 3.09 | 7.40 | 0.01535 |
| 12 | 24 | 13.1 | 3.42 | 7.80 | 0.00974 |
| 13 | 26 | 12.0 | 3.76 | 8.14 | 0.00510 |
| 14 | 28 | 11.9 | 4.17 | 8.49 | 0.00202 |
| 15 | 30 | 11.8 | 4.46 | 8.86 | 0.00096 |

| | |
|---|---|
| U_wind | Wind speed at hub height |
| Ti | Turbulence intensity |
| H | Significant wave height |
| Tp | Spectral peak period |
| Pro_occ | Probability of occurrence |

### 4.4 Calibration of soil-foundation models

#### 4.4.1 Calibration methodology

The calibration of all the four models is based on the linearized foundation stiffness used in Phase II of the comparison exercise IEA OC3 (Jonkman and Musial, 2010) to calibrate the linear soil-structure interaction models. The linearized foundation stiffness corresponds to the secant foundation stiffness calculated by Passon (2006) by means of API *p-y* curves (with the geotechnical code LPILE) for a horizontal load of 3.91 MN applied 31.87 m above mudline. The foundation stiffness of the four models evaluated in this study is calibrated as follows:

- Model 1 uses the API *p-y* curves calibrated in the comparison exercise IEA OC3 by Passon (2006).
- Model 2 and 3 are calibrated based on the linearized foundation stiffness calculated in the comparison exercise IEA OC3.
- The stiffness of Model 4 is based on a load-displacement curve obtained from finite element analyses (FEA). To be consistent with the calibration of the other models, the load-displacement curve from the finite element analyses was scaled to fit the secant stiffness used in the comparison exercise IEA OC3 at the load level defined in Passon (2006). The FEA were performed to establish a realistic non-linear behaviour.

In the following sections, the details of the calibration of each of the models are presented.

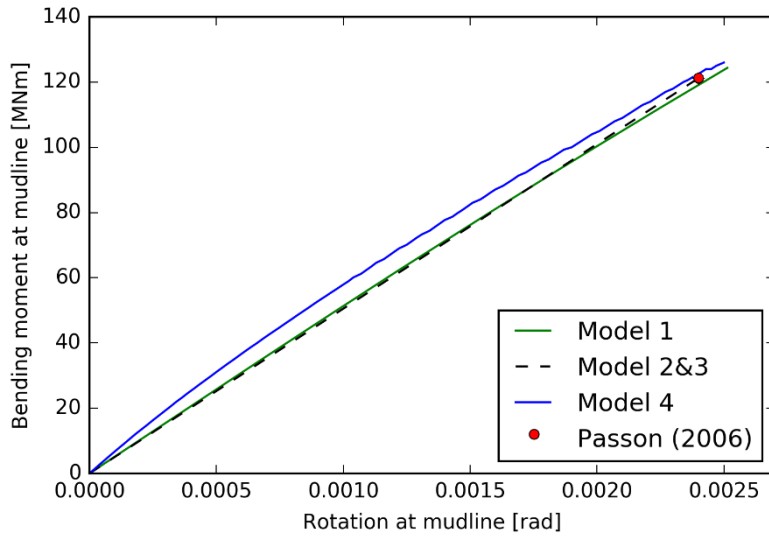

**Figure 11: Rotational stiffness characteristics for the different models.**

### 4.4.2    Calibration of Model 1

The p-y curves generated by Passon (2006) for OC3 Phase II, which follow the API sand model, have been used in this study with the purpose of benchmarking the study against industry practice. The reader is referred to Passon (2006) for more details. In the load region relevant for this study, the p-y curves show little non-linearity (Figure 11). A selection of p-y curves from the different soil layers is presented in Figure 12.

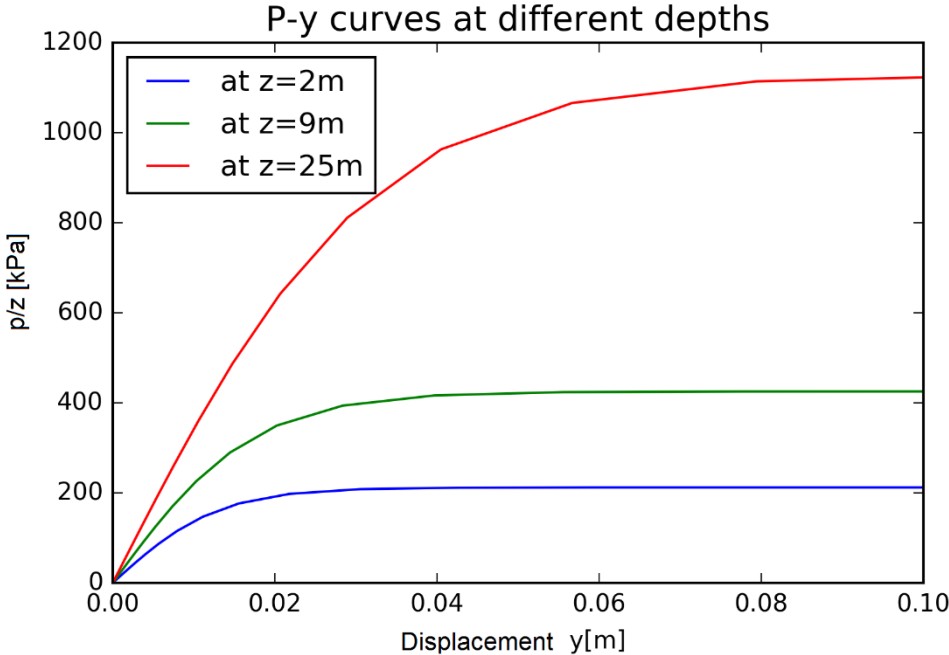

**Figure 12: Examples of p-y curves from the different soil layers.**

### 4.4.3    Calibration of Model 2

Parameters for the foundation stiffness matrix have been defined based on the coupled-springs model of OC3 Phase II (Jonkman and Musial, 2010). Simple soil-foundation models were calibrated for the project by Passon (2006). As stiffness coefficients were produced for a 2D system, the stiffness matrix has simply been extended to a 3D system, by using the same stiffness coefficients along both horizontal axes. By this approach, coupling effects between the two horizontal axes are neglected. This simplification is examined later in this paper.

Passon (2006) estimated the stiffness coefficients by calculating the secant pile stiffness at mudline at a given load level with the geotechnical code LPILE. In these analyses, the pile was modelled as a beam and the pile-soil interface and soil response were modelled as uncoupled lateral p-y springs. The secant stiffness was calculated for 1.5 times the ULS loads. This should be considered as a low stiffness estimate. The stiffness coefficients are given in Eq. (3)

$$
\begin{bmatrix}
k_{xx} & 0 & 0 & 0 & k_{x\beta} & 0 \\
0 & k_{yy} & 0 & k_{y\alpha} & 0 & 0 \\
0 & 0 & 0 & 0 & 0 & 0 \\
0 & k_{\alpha y} & 0 & k_{\alpha\alpha} & 0 & 0 \\
k_{\beta x} & 0 & 0 & 0 & k_{\beta\beta} & 0 \\
0 & 0 & 0 & 0 & 0 & 0
\end{bmatrix}
\qquad
\begin{aligned}
k_{xx} &= k_{yy} = 2.57481 \cdot 10^9 \quad \textit{[N/m]} \\
k_{\alpha\alpha} &= k_{\beta\beta} = 2.62912 \cdot 10^{11} \quad \textit{[Nm/rad]} \\
k_{x\beta} &= k_{\beta x} = -2.25325 \cdot 10^{10} \; \textit{[N/rad], [N]} \\
k_{y\alpha} &= k_{\alpha y} = 2.25325 \cdot 10^{10} \quad \textit{[N/rad], [N]}
\end{aligned}
\qquad (3)
$$

, where $x$ and $y$ are displacements in the horizontal plane, and $\alpha$ and $\beta$ are rotations around the corresponding axes. Wind and waves are aligned with the x-axis for all load cases in this paper.

### 4.4.4    Calibration of Model 3

Model 3 uses the same stiffness matrix as Model 2, giving it the same stiffness profile as Model 2. In addition, viscous rotational dampers have been included at the mudline, around both horizontal axes, to account for soil damping (represented in a damping matrix at the mudline node). The viscous dampers have been calibrated to give a foundation damping factor of approximately 1% near rated wind speed conditions (load case 6). This is considered reasonable and in line with studies from literature (Shirzadeh et al., 2013, Carswell et al., 2014). The foundation damping factor of 1 % expresses the hysteretic energy loss in the soil as a percentage of the total elastic strain energy of the soil. If the soil damping factor is expressed as the hysteretic energy loss in the soil as a percentage of the total strain energy of the complete OWT structure, the 1% damping value would decrease to 0.3%. The damping factor should not be confused with the damping ratio, as percentage of critical damping, also used throughout the paper.

To see how soil damping affects fatigue damage, two other calibrations for the damping coefficients have been chosen, giving foundation damping factors of 0.5% and 1.5% near rated conditions (load case 6). The calibrations for the rotational dampers are given in Table 2. The damping coefficients have been held constant for all load cases, as opposed to Model 4, where the inherent damping from hysteresis is amplitude-dependent.

**Table 2: Model 3 damping parameters**

| | $c_{\alpha\alpha}, c_{\beta\beta}$ [Nms/rad] | Foundation damping Factor (D)* | Foundation damping Ratio ($\xi_{fdn}$)* |
|---|---|---|---|
| Model 3a | 4.67e8 | ~0.5% | ~0.15% |
| Model 3b | 9.34e8 | ~1.0% | ~0.3% |
| Model 3c | 1.40e9 | ~1.5% | ~0.45% |

*Calculations are done for a natural frequency of 0.25 Hz

### 4.4.5    Calibration of Model 4

A finite element analysis of the soil-pile system was performed to obtain moment-rotation and horizontal load-displacement curves at the mudline. The analysis was performed with the geotechnical finite element software PLAXIS 3D with approximately 45 000 10-noded tetrahedral elements. Figure 13 illustrates the dimensions and the mesh refinement of the finite element model. Only half of the pile and the soil volume were modelled since both the geometry and the load acting on the pile are symmetric. A horizontal load of 1.955 MN was applied to half of the FE-model. The horizontal load – horizontal displacement curve at seabed was compared with a FE-model with significantly refined mesh and the discretization error shown to be less than 1% for the load range considered in the study. This indicated that the mesh discretization used in the study is sufficient.

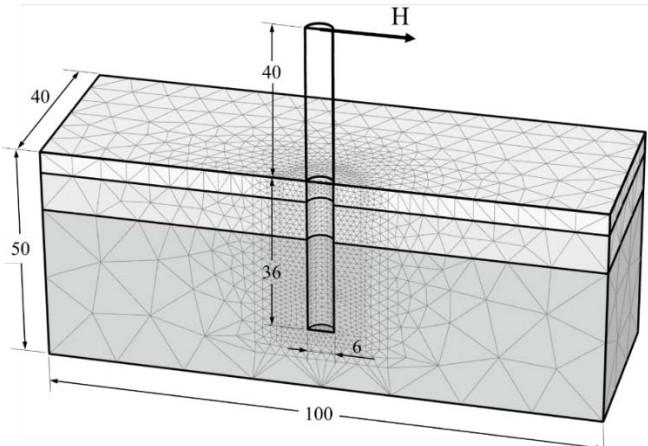

**Figure 13: Mesh and dimensions of the finite-element model.**

The Hardening Soil Small Strain constitutive model (Benz, 2007, Brinkgreve et al., 2013) was used to represent the sand behaviour. This constitutive model captures the very small strain soil stiffness and its non-linear dependency on the strain amplitude, and it is suitable for analyses of geotechnical structures in sand subjected to small-amplitude loading. Due to lack of soil test data, the parameters of the model were correlated from the relative density (RD) of the three sand layers based on the relations proposed by Brinkgreve et al. (2010). The relative densities of the sand layers were derived from the friction angle ($\phi'$) documented in Passon (2006) through the expression:

$$\phi' = 28 + RD/8 \tag{4}$$

Model 4 was calibrated by fitting the computed bending moment – rotation curve at mudline from finite element analyses, as illustrated in Figure 14. The results from Passon (2006), used in the calibration of Models 2 and 3, are included as a reference. In addition, the comparison between the computed bending moment – horizontal displacement curve (Figure 15) was used to determine the point of application of Model 4. The best fit was obtained when Model 4 was located 10.0 m below mudline. Figure 11 shows the stiffness of Model 4, compared with the other models. Seen relative to the other models, it gives a stiffer behaviour for low load levels, and softer behaviour for higher load levels.

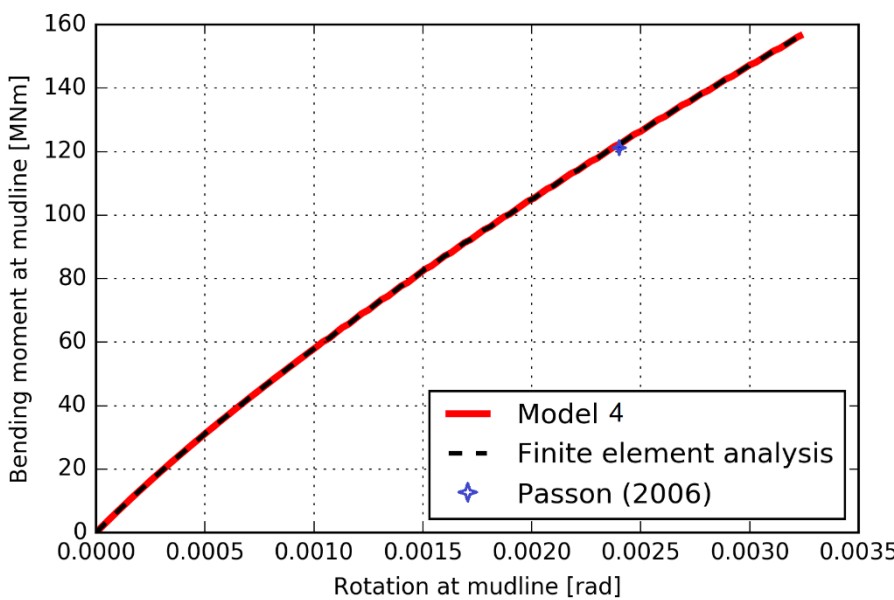

**Figure 14: Computed moment – rotation curve at mudline from finite element analyses and from the calibrated Model 4. The representative moment and rotation at mudline used in the calibration from Passon (2006) are included as a reference.**

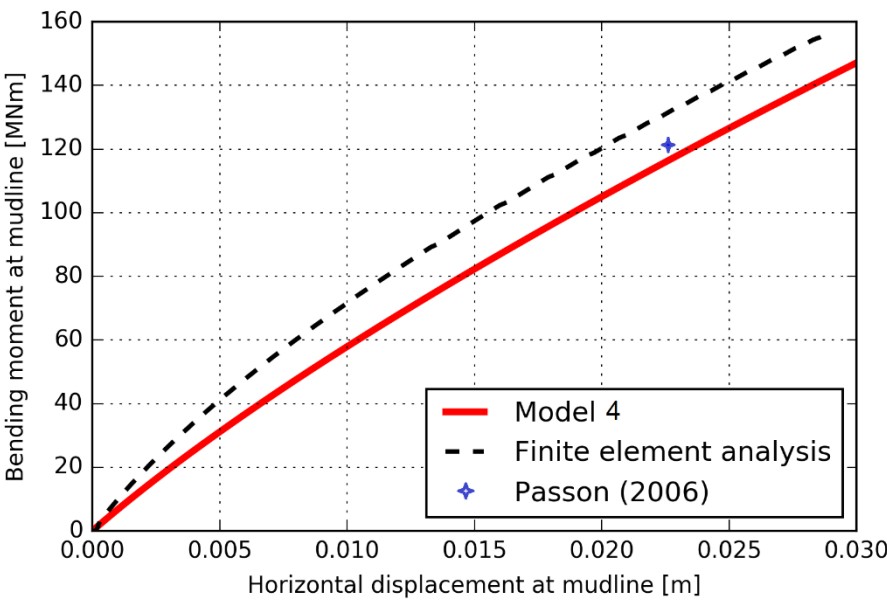

5    **Figure 15: Computed moment – horizontal displacement curve at mudline from finite element analyses and from the calibrated Model 4. The representative moment and rotation at mudline used in the calibration from Passon (2006) are included as a reference.**

## 4.5    Fatigue damage calculations

Fatigue damage can be evaluated in both the time- and frequency domain. For time domain simulations, the S-N curve methodology is widely used, and is briefly described below. Frequency domain simulations can also be performed using Dirlik's method. A comparison between these methods can be found in Ragan and Manuel (2007). More details on spectral methods for fatigue assessment can be found in Yeter et al. (2013) and Michalopoulos (2015).

In this study fatigue damage has been calculated by the S-N curve approach, using Palmgren-Miner's rule according to DNV standards (Det Norske Veritas, 2010). S-N curves gives the number of cycles before failure, for given stress ranges, $\Delta\sigma$. With variable stress ranges, linear cumulative damage is assumed, according to Palmgren-Miner's rule. The total damage at a given location is given by:

$$D = \sum_{i=1}^{k} \frac{n_i}{N_i} \tag{5}$$

, where all stress cycles are collected in $k$ number of stress blocks. $D$ is the accumulated fatigue damage (failure when $D$=1), $n_i$ is the number of stress cycles in block $i$, and $N_i$ is the number for cycles before failure for stress block $i$.

S-N curves for steel in air are used for calculations above the sea water line. At the mudline S-N curves for steel in seawater with cathodic protection has been used. S-N curve F3, from table 7-14 in the DNV standard DNV-OS-J101 (DNV, 2014) are used in fatigue calculations, as recommended for tubular girth welds. The curves are according to:

$$\log_{10} N = \log_{10} a - m \log_{10}\left(\Delta\sigma \left(\frac{t}{t_{ref}}\right)^k\right) \tag{6}$$

, where $N$ are the number of stress cycles before failure at stress range $\Delta\sigma$, $m$ is the negative slope of the logN-logS curve, $\log_{10} a$ is the intercept of the logN axis, $t_{ref}$ is a reference thickness, $t$ is the thickness through which the potential fatigue crack will grow, and $k$ is a thickness exponent. The S-N curve has different parameters, depending on the number of stress cycles. Parameter values used in this study are given in Table 3.

The duration of each load case is 1800 seconds. Results have been extrapolated to find the accumulated fatigue damage per year.

**Table 3: S-N curve parameters**

| | Air (Tower root) | | Seawater with cathodic protection (Mudline) | |
|---|---|---|---|---|
| | $N<10^7$ | $N>10^7$ | $N<10^6$ | $N>10^6$ |
| $\log_{10} a$ | 11.546 | 14.576 | 11.146 | 14.576 |
| $m$ | 3.0 | 5.0 | 3.0 | 5.0 |
| $k$ | 0.25 | 0.25 | 0.25 | 0.25 |
| $t_{ref}$ | 25mm | 25mm | 25mm | 25mm |

# 5    Analysis

## 5.1    Model characteristics from free vibration tests

A free vibration test was performed to identify the basic characteristics of the foundation models in terms of stiffness and damping. A forced displacement of 0.2 m at the tower top was applied and released, and then the tower was allowed to vibrate freely. This leads to some disturbances in the first cycles due to energy exchange between different modes, as can be seen in Figure 17.   The load amplitudes in the free vibration test are representative for load case 6 – 15. Damping characteristics of the models are presented in Figure 17. The eigen frequencies (based on the time between two consecutive peaks) are presented in Figure 18.

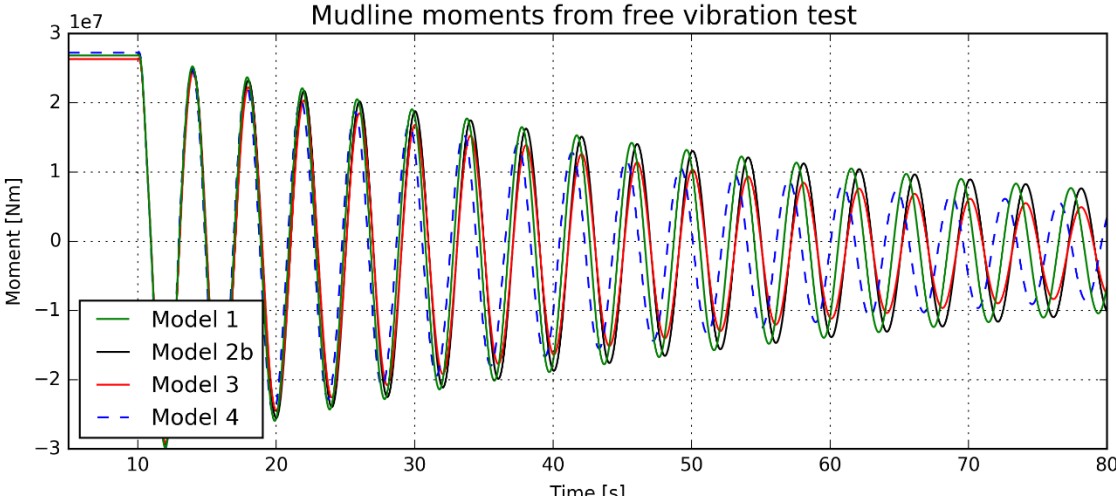

**Figure 16: Free vibration test with a tower top displacement of 0.2 m.**

Model 2 and 3 give the same 1$^{st}$ natural fore-aft frequency for the structure, as the mudline stiffness is the same. Model 1 and 4 shows stiffer behaviour, as a result of both the calibration methodology and load level. In Figure 17 and Figure 18 it can be seen how Model 4 gives lower damping and increased stiffness, as the loading amplitude decrease, which is more realistic behaviour compared to the other models.

The damping contribution from the different models is quantified by the global damping ratio, which is given as percentage of critical damping. It includes soil-, structural-, aerodynamic- and hydrodynamic damping. Only Model 3 and 4 account for soil damping. As damping sources other than soil damping are the same for all models, the differences are due the damping properties of the soil-foundation models. The soil damping factor for Model 3 is constantly 0.3%, and for Model 4 it varies between 0.05% - 0.3%.

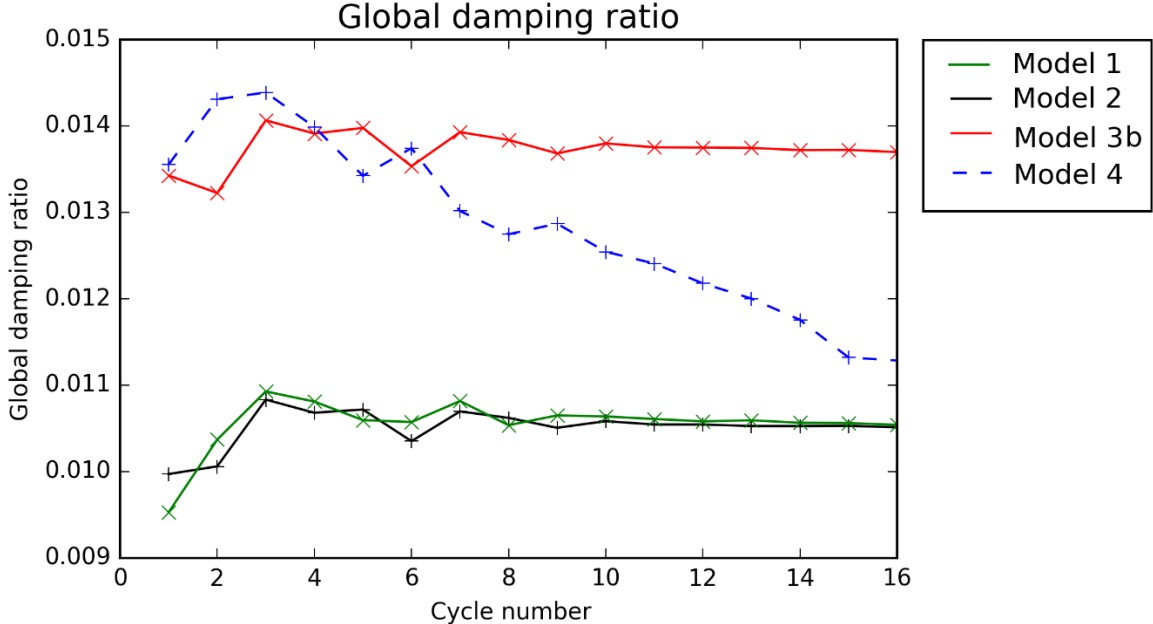

**Figure 17: Global damping ratio with the different foundation models.**

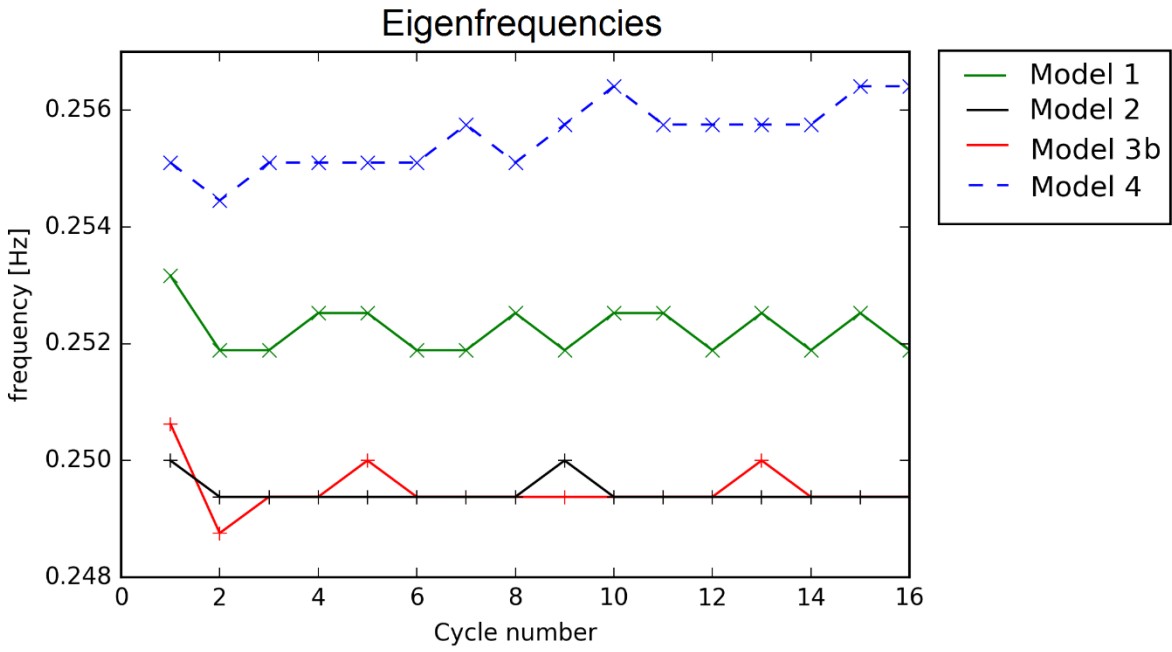

**Figure 18: Eigen frequencies for the foundation models.**

## 5.2 Foundation model impact on fatigue damage

The total accumulated fatigue damage per year is plotted in Figure 19, with the expected fatigue life time in years (y) at the top of each column. It can be seen that Model 2 gives the highest fatigue damage, and Model 4 the lowest, with a reduction of 22% relative to Model 2. The reduction is a consequence of a favourable increase in both stiffness and damping in Model 4 compared to Model 2. While a slightly larger stiffness in Models 2 and 3 could have moderated this conclusion, the results support the value of using non-linear hysteretic models that automatically account for the soil response in different load regimes. A comparison of Model 2 and 3 shows how soil damping by itself influence fatigue damage. Model 2, 3a, b and c are identical in their stiffness properties, therefore the differences in fatigue damage must be a consequence of the damping properties of the models. Table 2 show the damping properties of Model 3a, b and c, with foundation damping factors of 0.5%, 1% and 1.5%. Model 3b with a foundation damping factor of 1% reduces fatigue damage by 8% relative to Model 2.

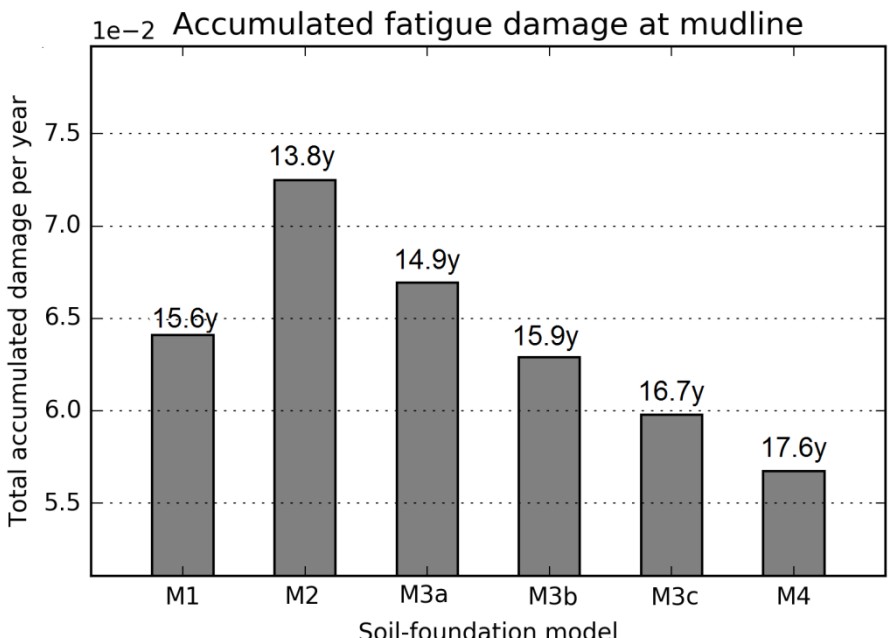

Figure 19: Accumulated fatigue damage at mudline.

Comparing the p-y curve approach (Model 1), with the linear elastic model (Model 2), it can be seen how the calibration methodology for the p-y curves is favourable to that of Model 2. Model 1 gives stiffer soil behaviour in the working load regions, being favourable in terms of fatigue damage. As a consequence of this, the softer Model 2 takes the natural frequency of the structure closer to wave frequencies than the stiffer Model 1, thereby increasing resonance effects.

Figure 20 presents the relative accumulated fatigue damage by load case. Results are normalized relative to the highest value. Each load case is also probability weighed, according to Table 1.

It can be seen how the foundation models have the highest impact on idling cases (LC 1 & 13-15). This has mainly two reasons: 1) LC 1 and LC 13-15 are load cases where the rotor is idling, and as a consequence aerodynamic damping is highly reduced. Thus the soil damping has a higher share of the total damping of the system. 2) Load amplitudes are higher. This leads to more damping from Model 4, while damping from Model 3a-c is unchanged.

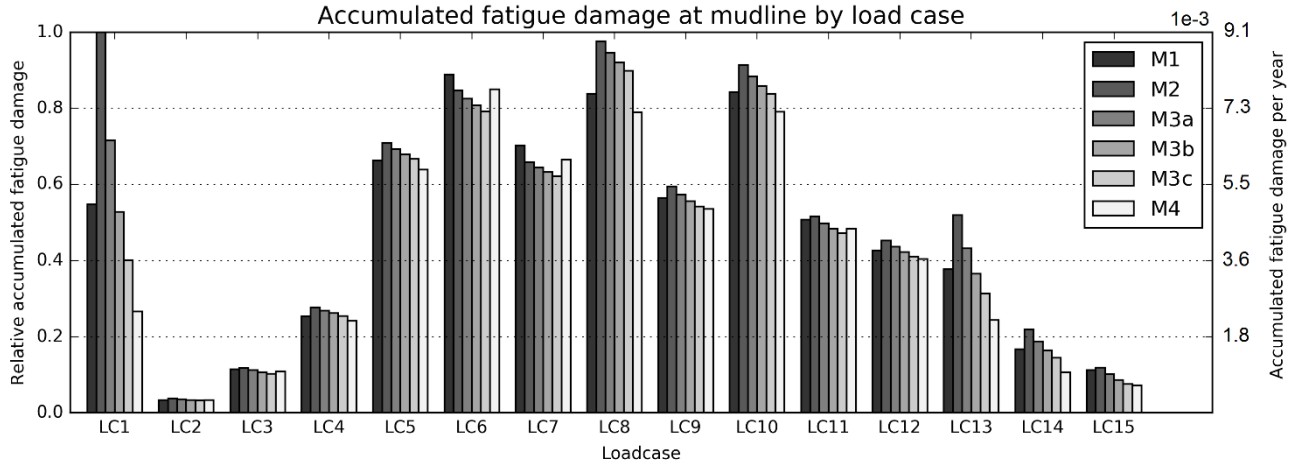

**Figure 20: Accumulated fatigue damage at mudline arranged by load case.**

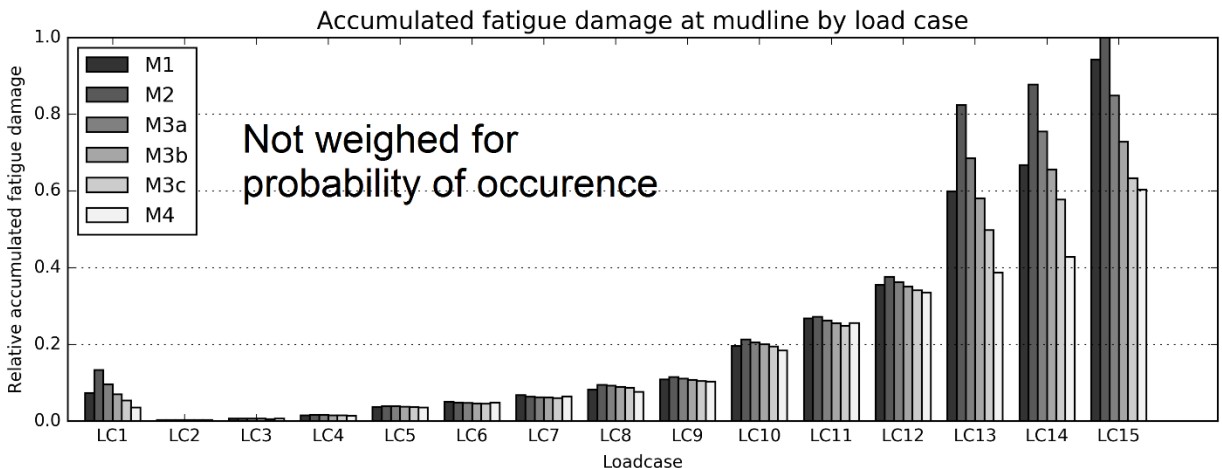

**Figure 21: Accumulated fatigue damage at mudline arranged by load case without probability of occurrence.**

Load case 1 has an impact on the total fatigue damage. This might seem counterintuitive, as wind and wave conditions are mild. However, with the rotor idle and limited aerodynamic damping, the tower is free to oscillate at its first natural frequency, leading to high load amplitudes at the mudline. Together with a high probability of occurrence, this gives a significant contribution to the total fatigue damage (8 % for Model4, approximately the same as LC4)

5    Figure 21 shows the fatigue presented again, but this time without taking the probability of occurrence into account, in other words the fatigue per unit time for the given environmental condition. Except for LC1, the trend is clear; increasing wind speeds and wave heights lead to higher fatigue damage. To illustrate the effect of aerodynamic damping, LC2 was run again (LC2b), this time with the rotor idling, and the blades pitched out of the wind. The mudline bending moment rms increased by a factor 2. Examples of mudline moment time series for LC2 and LC2b are shown in Figure 22.

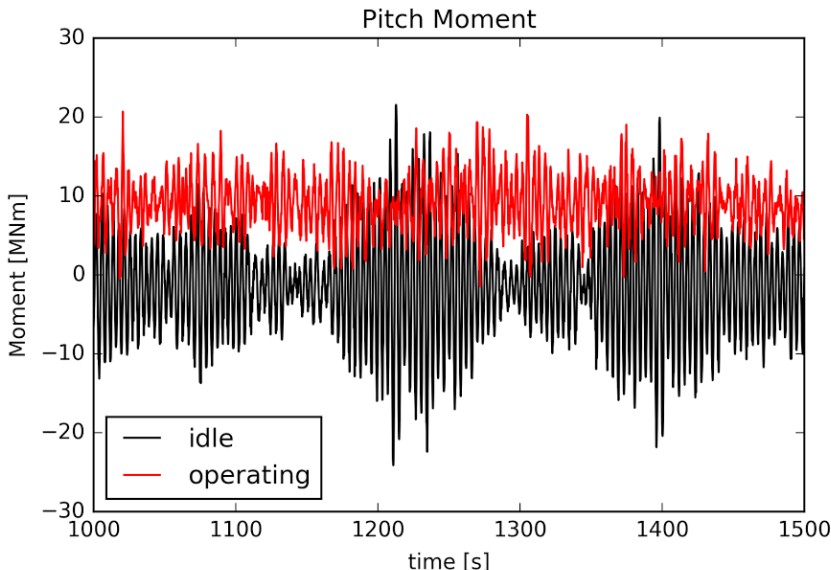

**Figure 22: Mudline pitch moment with an operating and idle rotor for load case 2.**

Fatigue damage calculations at the tower root (10 m above still water line) are given in Figure 23 and Figure 24. Absolute values are highly reduced, but the relative effect of the soil-foundation model shows to be even higher at this location. The
15    trends are similar to what was observed at the mudline.

Calculations were also done for the tower top and blade root. As the soil-foundation model had very little impact here (<1% change in fatigue damage), the results are not included in this paper. The differences in tower top motions due to foundation differences are not large enough to influence the rotor fatigue loads, driven by 3P tower blockage effects, 1P gravity loads and wind shear (for the blade roots) and turbulence. Relatively small changes in overall rotor loads, however,
20    will influence the bending moments and stresses at the mudline, due to the long moment arm.

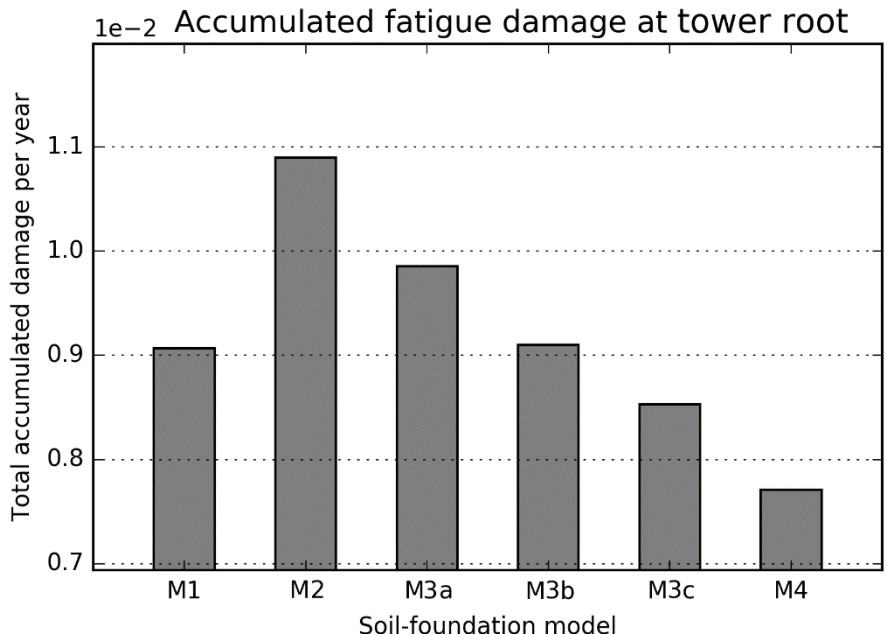

**Figure 23: Accumulated fatigue damage at tower root**

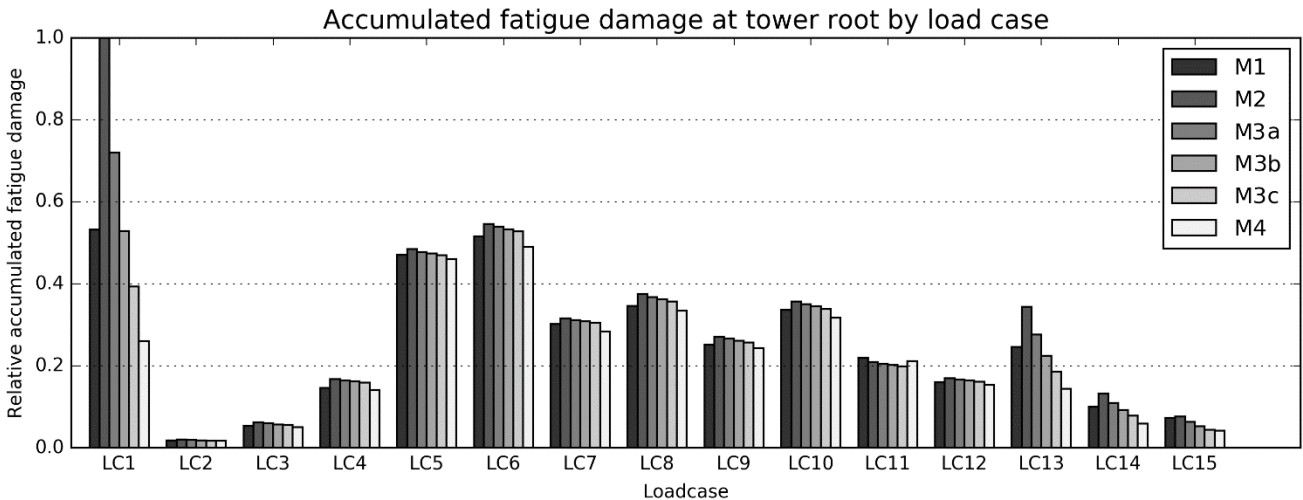

5    **Figure 24: Accumulated fatigue damage at tower root arranged by load case.**

### 5.3 Neglecting coupling between longitudinal and lateral motions for the soil models.

For non-linear models of soil response to pile motions, coupling terms do arise for large combined longitudinal and lateral motions. This has not been implemented in the current study; the models have simply been applied independently along the two horizontal axes. For all load cases, the waves and wind are aligned with the x-axis. The highest transversal motions relative to longitudinal motions are seen for LC15. Figure 25 shows the trajectory of the mudline node, with transversal motions up to about 2/3 of the longitudinal motions.

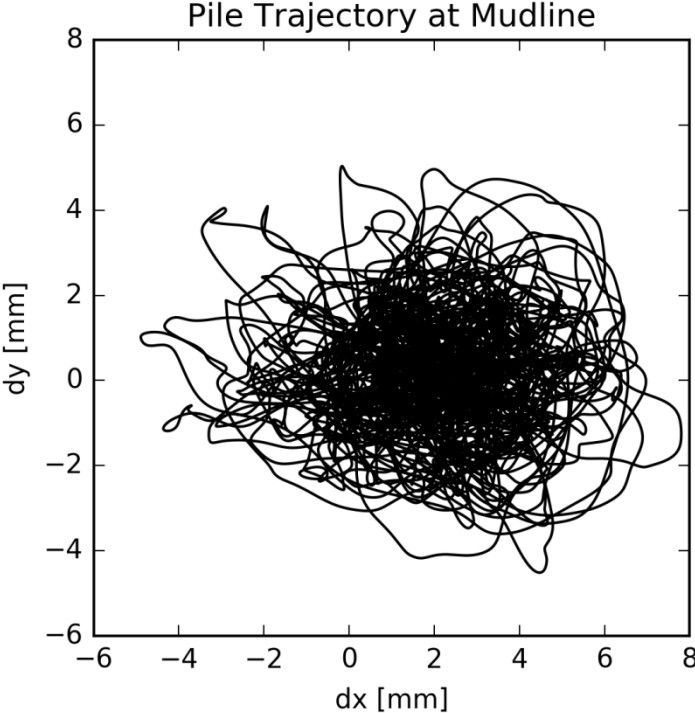

**Figure 25: Pile trajectory at the mudline**

To test the validity of ignoring the coupling between the horizontal motions, a run with wind, waves and rotor aligned with the x axis was compared with a corresponding run with wind, waves and rotor rotated at direction 45 degrees. The longitudinal (Roll) and transversal (Pitch) moment components are compared in Figure 26 and Figure 27. The changes are small, and the approximation seems to be acceptable for the cases in this work. For future cases with offsets between wind and wave directions, however, this assumption should be re-visited.

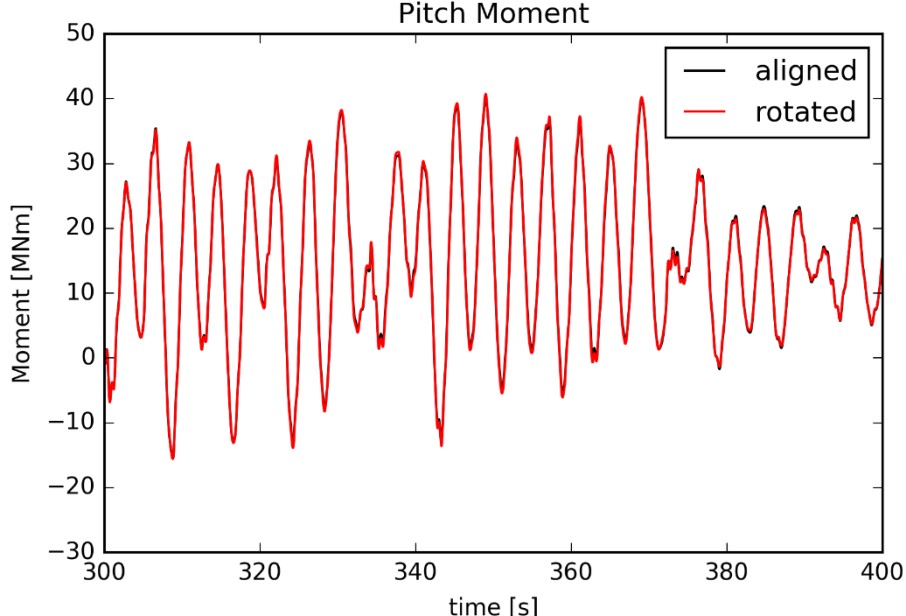

**Figure 26: Pitch moment for wind/wave direction 0 and 45 deg.**

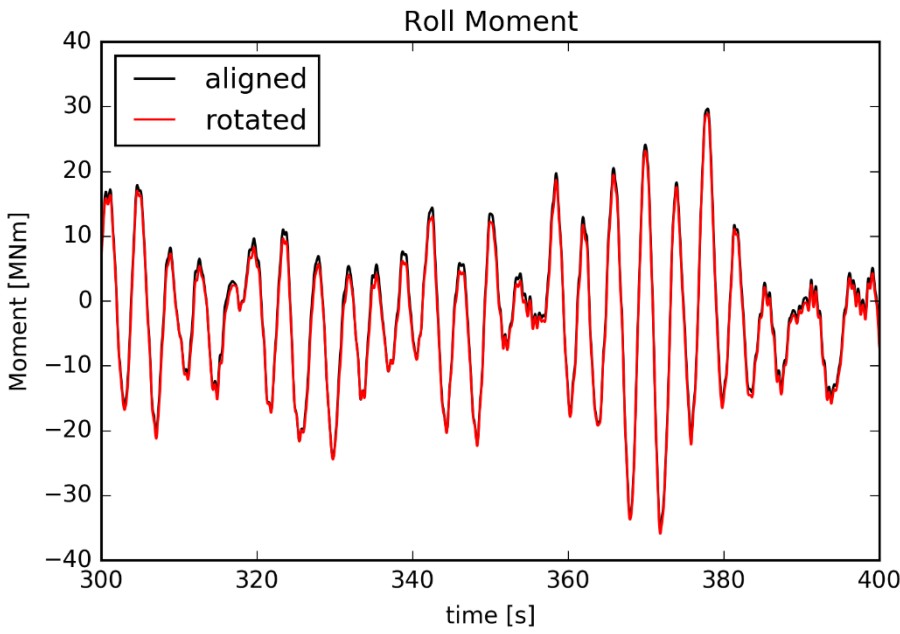

**Figure 27: Roll moment for wind/wave direction 0 and 45 deg.**

# 6   Conclusion and further work

The comparison of different soil-foundation models shows how both stiffness and damping properties influence the fatigue damage of an OWT with monopile foundation. This paper compares four soil-foundation models; the current industry standard (Model 1), a linear-elastic model (Model 2), a linear-elastic model that includes damping (Model 3) and a non-linear hysteretic model (Model 4).

Model 1 is the industry standard today. Non-linear springs (p-y curves) are attached at several levels below the mudline. The model represents the non-linear behaviour of the soil, but neglects soil damping. The calibration taken from the IEA OC3 project gives a slightly stiffer foundation at low load levels, compared with the linear-elastic models 2 and 3. A stiffer foundation increases the first tower natural frequency, leading to an increased margin to the wave excitation frequencies. Despite having no foundation damping, Model 1 gives about the same fatigue damage as Model 3b (linear-elastic with damping), due to the higher stiffness.

Model 2, a stiffness matrix adopted from the IEA OC3 project applied at the mudline, with different levels of damping (Model 3a, b and c), also applied at the mudline, demonstrate how increased foundation damping levels reduce fatigue damage for all load cases. This is because the first natural frequency of tower bending is relatively close to the wave frequencies, making small changes in damping important for the response. For the idle cases, with limited aerodynamic damping, the resonant response becomes even more sensitive to small changes in foundation damping. Compared to Model 2 (linear-elastic with no damping) the total fatigue damage was reduced by 13% when adding a foundation damping factor of 0.3% (Model 3b).

The non-linear hysteretic Model 4 reduces fatigue damage as a consequence of both its damping and stiffness properties. Model 4 is stiffer when the loading amplitude is small, which is true for the majority of operational time. This brings the natural frequency of the system away from the wave frequencies, resulting in less resonance effects. Model 4 includes damping as function of load amplitude due to hysteresis, giving more damping at high loads levels. Both effects reduce the load amplitude. Compared to Model 1 (p-y curves) and Model 2 (linear-elastic), the accumulated fatigue damage at mudline was reduced by 11% and 22% respectively.

The changes in fatigue damage due to the different foundation models in this study can to a large extent be explained in terms of how appropriate the stiffness and damping levels are for the load cases at hand. It can therefore be argued that Model 1 and 3, tuned to groups of load cases, would give similar results for fatigue damage as the more detailed Model 4. We nevertheless recommend Model 4, where the non-linear hysteretic behavior observed in piled foundations is accounted for. This gives stiffness and damping properties changing according to the different load levels, reducing the need for re-calibration for the different load cases.

This study, along with other studies of bottom-fixed offshore wind turbines, has brought the attention to idling cases, where the absence of aerodynamic damping and high probability of occurrence gives both a high contribution to the

total fatigue damage, and a high sensitivity to the foundation model. The study used a lumped wind/wave diagram, with both wind and waves acting in the same direction. This should be revisited in the continuation of this work, as cases with offsets between wind and wave directions could lead to relatively high excitation from the waves, and low aerodynamic damping in the direction of the waves. The details of the foundation model could here become even more important.

# 7    Author contributions

|  | Aasen | Page | Skjolden-Skau | Nygaard |
|---|---|---|---|---|
| **On the research idea and outline of the paper** | | | | |
| *Research idea* | | x | X | |
| *Structure of the research* | | x | X | x |
| *Outline of the paper* | | x | X | x |
| *Literature review* | x | x | | |
| **On the implementation of foundation models** | | | | |
| *Implementation of Models 1, 2 and 3 in 3DFloat* | | | | x |
| *Implementation of Model 4* | | | x | |
| *Implementation of the interface between 3DFloat and Model 4* | | | | x |
| **On the calibration of foundation models** | | | | |
| *Calibration of Models 2 and 3* | x | | | |
| *Calibration of Model 4* | | x | x | |
| *Calibration of Model 1* | | x | x | |
| **On the simulations** | | | | |
| *Building the simulation models in 3DFloat* | x | | | x |
| *Carrying out the simulations* | x | | | |
| *Extracting results from the simulations* | x | | | |
| **On analysing and discussing the results** | | | | |
| *Analysing the simulation results* | x | | | |
| *Discussion of the simulation results* | x | x | x | x |

# 8    Competing interests

The authors declare that they have no conflict of interest.

# 9    Acknowledgements

The financial support by the Norwegian Research Council through the project Reducing Cost of Offshore Wind by Integrated Structural and Geotechnical Design (REDWIN), Grant No. 243984, is gratefully acknowledged. The authors also want to acknowledge the support from Hans Petter Jostad, NGI (in the discussion of the initial research idea), Jörgen Johansson, NGI (in the definition of the structure of the research and in the calibration of Model 3), Jacobus Bernardus De Vaal, IFE (in setting up the 3Dfloat input) and Gudmund Reidar Eiksund,  NTNU (in reviewing the final document).

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
