# Peer review of "Effect of the Foundation Modelling on the Fatigue Lifetime of a Monopile-based Offshore Wind Turbine"

_Wind Energy Science, 2016_

## Referee Comment (RC1) · Anonymous Referee #1 · 9 Jan 2017

This paper studies the effects of different soil-foundation models on the fatigue damage of a OWT with monopile foundation. The results show that both stiffness and damping properties have a noticeable effect on the fatigue damage.

The comments are as follows.

1. Introduction

a) A paragraph reviewing methods used for modelling soil-solid interactions, such as p-y curve method and 3D (three-dimensional) FEA (finite element analysis) method, should be added.

b) A review on relevant studies, such as fatigue assessment of OWT (offshore wind

turbine monopile), should be added.

c) It would be appropriate to present the outline of the paper at the end of the introductory section, making the structure of the paper clear.

2. Section 3.1

Please give more details about how the wind and wave loads were applied on the monopile. Were they applied as point load or distributed load?

3. Section 4.1

Please give the thickness of the monopile used in this study.

4. Section 4.2

For soil profile, please present the p-y curves for three types of sands used in this study, i.e. loose, medium and dense sands.

5. Section 4.3

Please present some load calculation results of both wind and wave loads, e.g. the load calculation results for Load Cases 5 and 6.

6. Section 4.4.3

Please give more details about the FE model, such as types of elements, mesh size, displacement boundary conditions, the amplitude of the horizontal load H, and contact type between the soil and pile. Were mesh sensitivity exercises performed?

7. Section 4.5

Please justify why DNV F3 in-air S-N curve was chosen in this study. Please give details how the wind and wave load period were determined for the fatigue analysis.

8. Section 5.3

According to Figs. 17 and 19, LC1 (load case 1) has a high impact on the total fatigue

damage. This seems unreasonable, as the both wind and wave loads are relatively low. Authors state "with little aerodynamic damping, the tower is free to oscillate at its first natural frequency, leading to high load amplitudes at the mudline". Can authors compare the load calculation results for LC1 obtained from 3Dfloat against the results obtained from other aero-hydro-elastic codes, e.g. NREL FAST, to confirm this?

---

## Referee Comment (RC2) · M Muskulus (Referee) · 24 Jan 2017

General comments

The paper is a case study that compares four different foundation models in the context of load analysis for offshore wind turbines. One model is the classical p-y approach, two models are strong simplifications using springs at mudline, and one model implements a nonlinear stiffness characteristic with hysteresis. The models are compared with respect to fatigue damage in time domain simulations using a typical set of operational load cases, including idling.

The topic is highly relevant for offshore wind energy, and it is interesting to see a new

foundation model evaluated. The paper is quite readable, and the topic suits Wind Energy Science journal well.

I would like to recommend publication, but I am not convinced of the value of the paper in its present form. My main issue with the paper is that the comparison of the different foundation models is not based on any common principles or a well thought of methodology. To put it differently, the comparison is not "fair", and therefore little knowledge can be gained from the study. Apart from that, I found a number of issues with the presentation that should be fixed. All of this is explained in my detailed comments below.

Specific comments

1. Section 4.4.3 describes the calibration of Model 3. This was performed using finite element simulations of soil behavior. However, important details about these simulations are missing - the number of elements, for example, and how the convergence of the solution was assessed. Why should we consider the results of these analyses as representative of reality? What gives us this confidence? Moreover, why were the loads applied with an arm of 40 m, whereas the real turbine has the thrust load acting at 107.6 m height? What loads were used? Were the properties of the soil model determined for each load case separately?

2. Section 4.4.3: to continue from the previous item, why were the other foundation models not also fitted to the results of the finite element simulations? This would allow for a much more interesting comparison. As it is, the different models lead to different natural frequencies and damping values of the wind turbine system, and therefore of course to differences in fatigue damage. It is unclear how results can be compared at all, and how the new Model 3 should be assessed relative to the existing models. A clear methodology is missing here.

3. Conclusions: "The results clearly show that choosing an appropriate conceptual

foundation model can have significant positive effects" - The word "positive" implies that we are interested in lower damage in the results of simulations, which is not necessarily the same as being interested in more realistic and more accurate models of the underlying physical processes. I would strongly recommend to remove the word "positive". Similarly, Model 3 seems to result in higher damping at high load levels, which is again judged to be "positive". The first question is, is it realistic? If you can demonstrate sufficiently that the Model is more accurate than the other models (even when these are fit and used optimally), then it is possible to mention that it is a positive thing that the damage is also less than previously thought.

4. Conclusions: I am missing a real conclusion of the study. What are the recommendations with respect to the discussed foundation models? Which one should we use?

5. Section 2.3 discusses previous work on numerical modelling of foundation effects in offshore wind turbines. While a number of recent references are given, I miss a mention of Zaaijer et al ("Foundation modelling to assess dynamic behaviour of offshore wind turbines", Applied Ocean Research 2006) where similar simplified spring models were studied, albeit for smaller wind turbines. Also the work of Achmus et al (e.g. "Behaviour of monopile foundations under cyclic lateral load", Computers & Geotechnics 2009) is relevant in this context.

6. Figures 2-5: The illustrations of the different foundation models are more confusing than useful and should be revised. For example, Figure 2 shows a point representing a stiffness matrix, whereas Figure 3 shows an additional rotational spring with a damper. However, the model in Figure 2 also has a rotational spring (if I understood it correctly), so why is there no spring in Figure 2 also? Figure 5 visualizing the model with a nonlinear characteristic and hysteresis contains an illustration of a number of elastic-perfectly plastic springs combined in parallel.

However, the spring used in the model is a rotational spring and not a translational spring as the illustration suggests.

7. There is some confusion with respect to the pile length below mudline. Figure 4 tells the reader that it is 10 m, whereas on the next page a value of 9.5 m is mentioned. It is also unclear to the reader why this particular value was chosen and if the results are sensitive to this aspect of the model.

8. Section 3.2.3: "Masing's rule" should be explained to readers not familiar with it.

9. Section 3.2.4: It is unclear which degrees of freedom are affected by this model. In particular, is it a 1D or 2D model? It is also unclear how the bottom of the pile is treated. Typically this is modelled as being clamped to the ground, to guarantee suitable boundary conditions for finite element analyses, but this seems not to be the case here?

10. The order of the models is somewhat unnatural. The classical p-y approach comes last, whereas it should probably be the first model. The new nonlinear model occurs in between, whereas it should probably be the last model.

11. Environmental data were taken from a 25 m site, but the study assumes 20 m of water depth. Why?

12. Section 4.4.1: "coupling effects between the two horizontal axes are neglected ... since mainly in plane loads are consider, the simplification is considered to be acceptable" - Does this reasoning also apply to the (important) idling load cases?

13. Figure 10: What are the scales? What are the units?

14. Figures 12-13: There seems to a mistake with the scales and units. The x-axes are similar, but with an order of magnitude difference? The y-axis in Figure 13 should read "Horizontal displacement ..."? The figure is also not displaying a moment-displacement curve as reported in the caption, is it?

15. Section 4.5: "Parameter values used in this thesis ..."? If the results in the paper are based on a master thesis, this needs to be mentioned (e.g. in a footnote) and the thesis should be cited.

16. Section 4.5: "absolute values should be evaluated accordingly" - I assume that you want to say that "the reported values should be evaluated accordingly?" It is not true that "This will not influence relative values ...", the results will be skewed to a certain degree - you should explain your reasoning here in more detail.

17. Figure 14: What are the eigenfrequencies of the different wind turbine models? The results of the decay test shown suggest that there are differences between the different models? Then the difference in accumulated damage reported in Figure 16 can probably be explained by these frequency differences already. It is unclear what further effects the differences in foundation modelling have. In particular, it is not true then that "it can be seen how the nonlinear foundation stiffness influence fatigue damage".

18. Figure 17: Comparing the damage for the idling load cases, it is unclear which of the wind turbine models can move in the side-to-side direction and which cannot - it seems that some foundation models restrict the relevant degrees of freedom, for example, whereas others do not. This could already explain a large amount of the differences and should be discussed in more detail. Especially since "this gives a significant contribution to the total fatigue damage" - please report quantitatively how large this contribution is, instead of just saying that it is "significant", which has no objective meaning here.

19. Figure 15 and Section 5.2: What are Models 2a and 2b? These should be introduced in Section 3.2, where the methods are described, not in the results section.

20. Section 5.2: "On these locations rotor dynamics dominate the loading, which are

not considerably influenced by the soil-foundation response." - This is somewhat surprising, as I would expect differences in damping (from the soil) and the eigen-frequencies of the structure to influence these results. Why is this not the case?

Technical corrections (p=page, l=line)

- p2, l15: "This foundation results in"?

- p4, l16: "depending on load level"?

- p4, l21: "and are a function"?

- p6, l11: "Airy wave components"?

- p6, l16: Reference to a webpage should preferably be a footnote

- p6, l16: Remove "crude", as this method has been the state of the art for a long time - and with reasons.

- p6, l20: "Model 4 refers"?

- p7, l7: "moments typically dominate ... for OWT monopiles"?

- p7, l11: I assume that omega is an "angular velocity"?

- p10, l13: "unit weight"

- p10, l10: One should focus on describing the model used to describe the wind, not on the particular implementation in specific software.

- p10, l13: "Airy wave"

- Table 1: Seems to contain a typo. The first probability of occurrence should be "0.00671"

- p12, l8: "are considered"?

- p13, l4: "the foundation stiffness for Model 1 is independent of load level" - this is presumably also the case for Model 2?

- Figure 10: "Rotational stiffness as a function ..."?

- p13, l16: Use decimal points for numbers, not decimal commas.

- p15, l2: "are included as reference" - It should be stressed here that these were not used to calibrate Model 3, to avoid confusion.

- Figure 12: Is this the same data as in Figure 10?

- Figure 15: The symbols in the graphs are not explained in the legend

- p20, l21: "Calculations were also done for the tower top and blade root".

---

## Author Comment (AC1) · 21 Feb 2017

**Response to Anonymous Referee #1**

**Discussion Paper wes-2016-37**

Effect of the Foundation Modelling on the Fatigue Lifetime

of a Monopile-based Offshore Wind Turbine

*Steffen Aasen, Ana Page, Kristoffer Skjolden Skau, and Tor Anders Nygaard*

Dear reviewer,

We would like to thank you for spending time and effort in reviewing our paper. Your comments are much appreciated. We have replied to each of them in the text below and specified which actions will be taken in the paper.

Best regards,

Steffen Aasen, Ana Page, Kristoffer Skjolden Skau, and Tor Anders Nygaard

**1. Introduction**

**a) A paragraph reviewing methods used for modelling soil-solid interactions, such as p-y curve method and 3D (three-dimensional) FEA (finite element analysis) method, should be added.**

We will include a paragraph reviewing the methods used for soil-solid interaction in the introduction after line 18. Our suggestion reads as follows:

"Different approaches can be used to model the soil-foundation response for piles. Generally, they are divided into two groups: continuum approaches and subgrade reaction approaches. In continuum approaches, the soil is treated as a continuum material described by a constitutive relation. The problem of a pile embedded in a continuum material can be solved analytically if the soil is assumed to be a linear-elastic material (e.g. Poulos (1971)) or numerically if the soil is characterized by a more complex constitutive relation. Among the numerical methods, the boundary element method (e.g. Kaynia and Kausel (1982)) and the finite element method (e.g. Randolph (1981) or Andresen et al. (2010)) are the most widely used. In the subgrade reaction approaches, the soil response around the pile is described by a set of uncoupled individual horizontal springs, where the interaction between layers is only taken into account by the pile continuity. The springs relate the local lateral resistance, $p$, to the local lateral displacement of the pile, $y$, following a predefined function. Several $p$-$y$ functions can be found in the literature (e.g. Reese and Van Impe (2010)), however, the API (API 2011) $p$-$y$ curves are the most widely used."

**b) A review on relevant studies, such as fatigue assessment of OWT (offshore wind turbine monopile), should be added.**

A short note on fatigue assessment for offshore wind turbines (OWT's) will be added in the introduction as follows:

"The most widespread methods for fatigue estimations are time domain simulations with S-N curves and frequency domain calculations. A comparison of these methods can be found in Ragan and Manuel (2007). The industry standard for fatigue damage calculations is the time domain simulations with S-N curves described by DNV (Det Norske Veritas 2014). This methodology is used in this article.»

More references will be added in section 4.5 for details on spectral methods:

"Fatigue damage can be evaluated in both time- and frequency domain. For time domain simulations, the S-N curve methodology is widely used, and is briefly described below. Frequency domain simulations can also be performed using Dirlik's method (Dirlik 1985). A comparison between these methods can be found in Ragan and Manuel (2007). More details on spectral methods for fatigue assessment can be found in Yeter et al. (2013) and Michalopoulos (2015)».

**c) It would be appropriate to present the outline of the paper at the end of the introductory section, making the structure of the paper clear.**

We agree with the reviewer's suggestion and will add at paragraph at the end of the introductory section as follows:

"Following the introduction, Chapter 2 gives a review of observed foundation behavior, current foundation models for OWT monopiles, and relevant studies investigating effects of soil stiffness and damping. Chapter 3 presents the simulation software 3DFloat and the different soilfoundation models studied in this paper. Chapter 4 presents the OWT structure, the soil profile and the environmental conditions that has been applied in simulations. The calibration of each soil-foundation model, together with our methodology for fatigue damage calculations is also included at the end of Chapter 4. Chapter 5 presents the results from the analysis that has been carried out, followed by a conclusion with suggestions for further work in Chapter 6."

**2. Section 3.1**

**Please give more details about how the wind and wave loads were applied on the monopile. Were they applied as point load or distributed load?**

The wind and wave loads are distributed on the structure. Forces per unit length are integrated with the interpolation functions used in the Galerkin formulation of the Finite-Element-Method. The distributed forces are thereby lumped to consistent nodal loads (forces and moments), applied to the nodes connecting the elements. The forces on the wet elements of the pile below the instantaneous wave surface are computed with Morison's equation. The quadratic drag forces on the tower above the instantaneous wave surface are computed from the turbulent wind. The wind turbine distributed blade loads are computed from Blade Element Momentum theory, taking into account the elastic deformation of the structure.

We will merge this information into section 3.1.

**3. Section 4.1**

**Please give the thickness of the monopile used in this study.**

We have modified 'Figure 9: Soil profile and pile dimensions' to include it, as shown in Figure 1.

[Figure]

*Figure 1: Soil profile and pile dimensions*

**4. Section 4.2**

**For soil profile, please present the p-y curves for three types of sands used in this study, i.e. loose, medium and dense sands.**

Following the reviewer's suggestion, we have plotted three *p-y* curves, one for each of the soil layers considered in this study. The blue curve at z = 2 m is representative for the top soil layer,

which extends from seabed to z = 5 m; the green curve is representative for the middle layer, which extends from z = 5 m to z = 14 m; and the red curve is representative for the bottom sand layer, which starts from z = 14 m. The vertical axis is normalized by the depth of the *p-y* curve considered.

This figure will be included in section 4.4.4., where the calibration of the *p-y* curves for the soil profile considered is presented.

[Figure]

*Figure 2: Representative p-y curves of each of the soil layers considered in this study. The vertical axis s normalized by the depth at which the p-y curve is calculated.*

**5. Section 4.3**

**Please present some load calculation results of both wind and wave loads, e.g. the load calculation results for Load Cases 5 and 6.**

We will include detailed results for the load cases (LC) 5, 6 and 1 (idle). LC 1 will be included to examine the significant contribution to fatigue, despite the mild wind and wave conditions for this case.

Figure 3 gives the tower top force in the wind direction, representing forces from the wind turbine (with inertial forces). Additional simulations must be run to extract wave forces, and will be included in the revised version of the paper.

[Figure]

Figure 3:Tower top force along x-axis (in wind direction)

**6. Section 4.4.3**

**Please give more details about the FE model, such as types of elements, mesh size, displacement boundary conditions, the amplitude of the horizontal load H, and contact type between the soil and pile. Were mesh sensitivity exercises performed?**

Thank you for this comment, this information should have been included in the text.

The commercial finite element software PLAXIS 3D AE was used to perform the analyses with 10-noded tetrahedral elements. Table 1 lists the model dimensions based on the axis shown in Figure 4. Only half of the pile and the soil volume were modelled since both the geometry and the load acting on the pile are symmetric. A horizontal load of 1.955 MN was applied to half of the FE-model. This horizontal load is the same one used by Passon (2006) to calibrate the elastic stiffness matrix at seabed in Phase II of the comparison exercise OC3 (Jonkman and Musial 2010).

*Table 1    Finite Element model dimensions*

| $x_{max}$ | $x_{min}$ | $y_{max}$ | $y_{min}$ | $z_{max}$ | $z_{min}$ |
|-----------|-----------|-----------|-----------|-----------|-----------|
| m | m | m | m | m | m |
| 63 | -63 | 42 | 0 | 0 | -50 |

The displacements at bottom boundary ($z_{min}$) are fixed, while roller boundaries are applied on the model sides ($x_{max}$, $x_{min}$, $y_{max}$, $y_{min}$). The mesh has 45 711 soil elements and 66 868 nodes. The average element size of the whole model is about 3 m, but significantly refined around the pile (approximately 1.3m). Figure 4 shows the mesh discretization.

Full contact was assumed between the pile and the soil.

Yes, a mesh sensitivity study was performed to assure that the mesh discretization was enough. In the mesh sensitivity study, two mesh discretization were compared:

1. The mesh discretization used in the paper, with 45 711 elements and 66 868 nodes, shown in Figure 4.
2. The finest mesh discretization that was possible to achieve in PLAXIS 3D, with 907 570 elements and 1 238 593 nodes, shown in Figure 5.

Number of elements: 45 711
Number of nodes: 66 868

[Figure]

Average element size: 1.3 m

*Figure 4: Mesh of the finite element model*

Number of elements: 907 570
Number of nodes: 1 238 593

[Figure]

Average element size: 0.5 m

*Figure 5: Mesh discretization used in the mesh sensitivity study*

The horizontal load – horizontal displacement curves at seabed calculated with the two mesh discretizations are compared in Figure 6. The almost perfect coincidence between the two curves demonstrates that the discretization in the paper is sufficient.

The figures showing the comparison will not be included in the paper. However, we will include in page 14 around line 6:

"The horizontal load – horizontal displacement curve at seabed was compared with a FE-model with significantly refined mesh and the discretization error shown to be less than 1 % for the load range considered in the study."

[Figure]

*Figure 6: Horizontal load – horizontal displacement curves at seabed calculated with the two mesh discretizations*

**7. Section 4.5**

**Please justify why DNV F3 in-air S-N curve was chosen in this study. Please give details how the wind and wave load period were determined for the fatigue analysis.**

Thank you for noticing this. The DNV F3 in-air S-N curve was chosen for simplicity. New calculations will be done at the mudline by using "DNV F3 in seawater with cathodic protection", as this position is exposed to seawater. This is the recommended S-N curve in DNV OS-J101 for tubular girth welds (Det Norske Veritas 2014).

Wind and wave periods are chosen from the Upwind Design Basis (Fischer et al. 2010), giving a lumped scatter diagram for wind and waves. The lumped scatter diagram should represent a possible shallow water site for monopile installation in the Dutch North Sea. This is further explained in section 4.3 of the paper. The lumping has been done according to methodology described by Kühn (2001).

**8. Section 5.3**

**According to Figs. 17 and 19, LC1 (load case 1) has a high impact on the total fatigue damage. This seems unreasonable, as the both wind and wave loads are relatively low. Authors state "with little aerodynamic damping, the tower is free to oscillate at its first**

**natural frequency, leading to high load amplitudes at the mudline". Can authors compare the load calculation results for LC1 obtained from 3Dfloat against the results obtained from other aero-hydro-elastic codes, e.g. NREL FAST, to confirm this?**

We think it is a good idea to examine this case closer. It is well known that the idle cases contribute significantly to fatigue damage for monopiles due to the low damping when the rotor is idle. The results are, however, sensitive to structural damping, the damping in the soil, the excitation around the tower eigen frequency, the damping inherent in the hydrodynamic and aerodynamic load models, and the nonlinearities and frequency content of the excitation from the quadratic drag models. We have neglected hydrodynamic radiation damping so far based on the literature review, but this can be included in the revised paper. The OC3 monopile has been studied with FAST and several other models in the IEA OC3 project, and we are going to invite colleagues to revisit the OC3 monopile for LC 1, 5 and 6.

**References**

Andresen, L., H. Petter Jostad and K. H. Andersen (2010). "Finite element analyses applied in design of foundations and anchors for offshore structures." International Journal of Geomechanics **11**(6): 417-430.

API (2011). Recommended Practice for Planning, Designing and Constructing Fixed Offshore Platforms - Working Stress Design, American Petroleum Institute.

Det Norske Veritas (2014). Design of offshore wind turbine structures, Det Norske Veritas**:** 238.

Dirlik, T. (1985). Application of computers in fatigue analysis, University of Warwick.

Fischer, T., W. De Vries and B. Schmidt (2010). UpWind Design Basis (WP4: Offshore foundations and support structures), Upwind.

Jonkman, J. and W. Musial (2010). Offshore code comparison collaboration (OC3) for IEA task 23 offshore wind technology and deployment**:** 275-3000.

Kaynia, A. and E. Kausel (1982). Dynamic behavior of pile groups. 2nd Int. Conf. on Numerical Methods in Offshore Piling, Austin, Texas.

Kühn, M. J. (2001). Dynamics and design optimisation of offshore wind energy conversion systems, TU Delft, Delft University of Technology.

Michalopoulos, M. (2015). Simplified fatigue assessment of offshore wind support structures accounting for variations in a farm, EWEA European Wind Energy Association.

Passon, P. (2006). "Memorandum: derivation and description of the soil-pile-interaction models." IEA-Annex XXIIII Subtask **2**.

Poulos, H. G. (1971). "Behavior of Laterally Loaded Piles: I-Single Piles." Journal of the Soil Mechanics and Foundations Division **97**(5): 711-731.

Ragan, P. and L. Manuel (2007). "Comparing estimates of wind turbine fatigue loads using time-domain and spectral methods." Wind engineering **31**(2): 83-99.

Randolph, M. F. (1981). "The response of flexible piles to lateral loading." Geotechnique **31**(2): 247-259.

Reese, L. C. and W. F. Van Impe (2010). Single piles and pile groups under lateral loading, CRC Press.

Yeter, B., Y. Garbatov and C. G. Soares (2013). "Spectral fatigue assessment of an offshore wind turbine structure under wave and wind loading." Developments in Maritime Transportation and Exploitation of Sea Resources: 425-433.

---

## Author Comment (AC2) · 21 Feb 2017

**Response to Referee #2**

**Discussion Paper wes-2016-37**

Effect of the Foundation Modelling on the Fatigue Lifetime

of a Monopile-based Offshore Wind Turbine

*Steffen Aasen, Ana Page, Kristoffer Skjolden Skau, and Tor Anders Nygaard*

Dear reviewer,

We would like to thank you for spending time and effort in reviewing our paper. Your comments are much appreciated. We have replied to each of them in the text below and specified which actions will be taken in the paper.

Best regards,

Steffen Aasen, Ana Page, Kristoffer Skjolden Skau, and Tor Anders Nygaard

**General comments**

**The paper is a case study that compares four different foundation models in the context of load analysis for offshore wind turbines. One model is the classical p-y approach, two models are strong simplifications using springs at mudline, and one model implements a nonlinear stiffness characteristic with hysteresis. The models are compared with respect to fatigue damage in time domain simulations using a typical set of operational load cases, including idling.**

**The topic is highly relevant for offshore wind energy, and it is interesting to see a new foundation model evaluated. The paper is quite readable, and the topic suits Wind Energy Science journal well.**

**I would like to recommend publication, but I am not convinced of the value of the paper in its present form. My main issue with the paper is that the comparison of the different foundation models is not based on any common principles or a well thought of methodology. To put it differently, the comparison is not "fair", and therefore little knowledge can be gained from the study. Apart from that, I found a number of issues with the presentation that should be fixed. All of this is explained in my detailed comments below.**

Thank you for the thoughtful and detailed review. We are glad to hear that you find the topic of the paper relevant for the Wind Energy Science journal. We have addressed the methodology and principle for comparison of foundation models in our replies to the specific comments, and the paper will be modified accordingly to better explain the overall approach for the comparison.

In short, the methodology for comparison is as follows:

1. All models are calibrated once
2. The models are calibrated at a load level as in the IEA OC3 project (Passon, 2006)
3. The methodology, in our opinion, is consistent with industry practice and the previous work on this wind turbine in the IEA OC3 project

**Specific comments**

**1. Section 4.4.3 describes the calibration of Model 3. This was performed using finite element simulations of soil behavior. However, important details about these simulations are missing - the number of elements, for example, and how the convergence of the solution was assessed.**

Thank you for this comment. We will include more details in the paper in section 4.4.3 of the paper (page 14 line 6).

The commercial finite element software PLAXIS 3D AE was employed to perform the analyses with 10-noded tetrahedral elements. A mesh with 45 711 soil elements and 66 868 nodes - shown in Figure 1 - was used. The convergence of the solution was assessed by performing a sensitivity study where two mesh discretizations were compared:

1. The mesh discretization used in the paper, with 45 711 elements and 66 868 nodes, shown in Figure 1.
2. A very fine mesh discretization with 907 570 elements and 1 238 593 nodes, shown in Figure 2.

The horizontal load – horizontal displacement curves at seabed calculated with the two mesh discretization are compared in Figure 3. The almost perfect coincidence between the two curves demonstrate that the mesh discretization used in the study is sufficient.

[Figure]

*Figure 1: Mesh of the finite element model*

[Figure]

Number of elements: 907 570
Number of nodes: 1 238 593

Average element size: 0,5 m

*Figure 2: Very fine mesh discretization used in the mesh sensitivity study*

[Figure]

+ 907 570 elements
— 45 711 elements

*Figure 3: Horizontal load – horizontal displacement curves at seabed calculated with the two mesh discretizations*

**Why should we consider the results of these analyses as representative of reality? What gives us this confidence?**

Finite element analyses in combination with an appropriate soil model is considered to be significantly more realistic than *p-y* curves. This is acknowledged by the geotechnical community and discussed among others in Lesny and Wiemann (2006), Schroeder et al. (2015) and Page et al. (2016), and is being accepted as the alternative approach by the industry. The soil behavior in the FEA reported in the paper is described with the Hardening Soil Small Strain model (Benz 2007), a model developed to describe the behavior of sands from small-strains to large strains.

The model has been validated against element tests and boundary value problems, providing good agreement to the measurements (Benz 2007, Sheil and McCabe 2016).

**Moreover, why were the loads applied with an arm of 40 m, whereas the real turbine has the thrust load acting at 107.6 m height? What loads were used? Were the properties of the soil model determined for each load case separately?**

Thank you for pointing out this. Indeed, the load level used should have been reported in the paper. A horizontal load of 1.955 MN was applied to half of the FE-model, which is equivalent to a horizontal load of 3.91 MN applied to the full model. This load was taken from the code comparison exercise OC3 (Passon 2006); where it was used to calibrate the linear soil-structure interaction models used in Phase II.

The load arm of 40 m was selected based on: (1) the evaluation of H/M-ratios in simulations and (2) the load arm used in the calibration of foundation stiffness used in the code comparison exercise OC3 (Passon 2006). The M/H ratio computed in Load Case 13 (LC13), where the offshore wind turbine is subjected to a turbulent wind of 26 m/s, is plotted in Figure 4 for a 30 s time window. The figure shows that the load arm varies but has a rough average between 20 – 50 m. Variation in H/M ratio within this range has little effect on the stiffness. A load arm of 31.87 m was assumed in the calibration of the OC3 linear soil-structure interaction models.

Concerning the last comment, no, the soil properties for the Hardening Soil Small Strain were not determined for each load case separately because the effect of long-term cyclic loading was not included in this study.

[Figure]

*Figure 4: Load arm (M/H) as a function of time from LC13, where a turbulent wind of 26 m/s was applied.*

**2. Section 4.4.3: to continue from the previous item, why were the other foundation models not also fitted to the results of the finite element simulations? This would allow for a much more interesting comparison. As it is, the different models lead to different natural frequencies and damping values of the wind turbine system, and therefore of course to differences in fatigue damage. It is unclear how results can be compared at all, and how the new Model 3 should be assessed relative to the existing models. A clear methodology is missing here.**

Thank you for this comment. We realize that we should have explained the logic behind our calibration methodology in a clearer manner. A section will be added within Section 4.4 with the title: "Calibration methodology". The calibration of all the four models is based on the secant stiffness found in the calibration of the foundation models in the comparison exercise IEA OC3. Figure 5 shows the moment vs. rotation at mudline for all the models during loading, unloading, and reloading, and the OC3 calibration.

The calibration of Model 3 is based on a load-displacement curve obtained from finite element analyses (FEA). To be consistent with the other three models, the load-displacement curve from the finite element analyses was scaled to fit the secant stiffness calculated by Passon. The FEA were performed to establish a realistic non-linear behavior. The FEA results (without scaling) are considered to provide a more "realistic stiffness" but it was chosen to scale the result to make the stiffness in the four models consistent. The reviewer argues that the consistency between models should have been related to the initial stiffness rather than the secant stiffness. We are not opposing such an approach, but since both Models 3 and 4 are non-linear, any load-displacement curve will deviate from the linear.

All the models could have been fitted to the results of the finite element simulations, however they would still have led to different natural frequencies of the wind turbine system depending on the load level. Figure 5b shows a comparison between the rotational tangent stiffness, which governs the first natural frequency, vs. the overturning moment from all the models. The tangent rotational stiffness of Model 3 can be both larger and smaller than the tangent rotational stiffness of Models 1, 2 and 4, depending in which part of the loading, unloading or reloading path we compare it. In the unloading and reloading curves, Model 3 gives almost the same tangent rotational stiffness as Models 1 and 2 for small moments (around the M=0 axis).

If Model 4 would have been calibrated to the results from finite element analyses (Figure 5c), then the tangent rotational stiffness for small moments would have been too large in comparison with the stiffness from Models 1, 2 and 3. In addition, the purpose of comparing Model 4 calibrated with API *p-y* curves was to benchmark the study against industry practice.

a)

[Figure]

b)

[Figure]

c)

*Figure 5: Stiffness of the different models during loading (points A to B), unloading (points B to D through C), and reloading (points D to B through E): (a) Moment vs. rotation at mudline for all the models; (b) Rotational tangent stiffness vs. moment at mudline for all the models; (c) Rotational tangent stiffness vs. moment at mudline for Model 3, Model 4 calibrated with API p-y curves, and Model 4 calibrated with the results from finite element analyses.*

**3. Conclusions: "The results clearly show that choosing an appropriate conceptual foundation model can have significant positive effects" - The word "positive" implies that we are interested in lower damage in the results of simulations, which is not necessarily the same as being interested in more realistic and more accurate models of the underlying physical processes. I would strongly recommend to remove the word "positive".**

We appreciate this point. We will remove the word "positive" and reformulate the sentence as follows: "The results show that choosing an appropriate conceptual foundation model, where the non-linear hysteretic behavior observed in piled foundations is accounted for, reduces the fatigue damage close to the foundation".

**Similarly, Model 3 seems to result in higher damping at high load levels, which is again judged to be "positive". The first question is, is it realistic? If you can demonstrate sufficiently that the Model is more accurate than the other models (even when these are fit and used optimally), then it is possible to mention that it is a positive thing that the damage is also less than previously thought.**

We will remove the word positive and reformulate the sentences as follows: "The nonlinear rotational model (Model 3) reduces the fatigue damage as a consequence of both its damping and stiffness properties".

The study considers a virtual case, and the word "realistic" should probably be used with care. However, the soil has a non-linear hysteretic damping nature which increases with the cyclic load amplitude (Darendeli 2001). Hysteretic load displacement loops can be observed in cyclic large- and small-scale pile tests and in centrifuge tests (Rosquoët et al. 2004).

The other models cannot be fitted optimally, because their model formulations are unrealistic compared to the observed soil and pile behaviour. However, they can be fitted to give a similar average response, as it has been done in this study (see Figure 5).

**4. Conclusions: I am missing a real conclusion of the study. What are the recommendations with respect to the discussed foundation models? Which one should we use?**

Model 3 is recommended, because it is more physically realistic without adding much computational effort. It includes the foundation behavior observed in pile tests: non-linear stiffness, different stiffness during loading and reloading and hysteretic damping, which increases with increasing cyclic load (and displacement) amplitudes. The model can also be calibrated to specific soil condition since it is based on FEA, which can model the effect of different soil layers and more complex soil behavior. We will include this more explicitly in the conclusions section.

**5. Section 2.3 discusses previous work on numerical modelling of foundation effects in offshore wind turbines. While a number of recent references are given, I miss a mention of Zaaijer et al ("Foundation modelling to assess dynamic behaviour of offshore wind turbines", Applied Ocean Research 2006) where similar simplified spring models were studied, albeit for smaller wind turbines. Also the work of Achmus et al (e.g. "Behaviour of monopile foundations under cyclic lateral load", Computers & Geotechnics 2009) is relevant in this context.**

Thank you for the comment, we will include the reference from Zaaijer (2006).

The paper from Achmus et al. (2009) is relevant while discussing the long-term behaviour of monopiles under large lateral cyclic loads, where accumulated pile displacements occur. However, in this paper, we have focused on relatively small fatigue loads, where the accumulated displacements are negligible. This makes the reference from Achmus et al. (2009) less relevant.

**6. Figures 2-5: The illustrations of the different foundation models are more confusing than useful and should be revised. For example, Figure 2 shows a point representing a stiffness matrix, whereas Figure 3 shows an additional rotational spring with a damper. However, the model in Figure 2 also has a rotational spring (if I understood it correctly), so why is there no spring in Figure 2 also? Figure 5 visualizing the model with a nonlinear characteristic and hysteresis contains an illustration of a number of elastic-perfectly plastic springs combined in parallel. However, the spring used in the model is a rotational spring and not a translational spring as the illustration suggests.**

Thank you for this comment. The stiffness matrix is sketched in both Figures 2 and 3 by a point, which represents both translational- and rotational linear elastic "springs". In Figure 3, the added element is representing a rotational dashpot, and not a rotational spring. We realize that we should have illustrated the foundation models in a clearer manner. We will update both the text and Figures 2 and 3.

The springs from the nonlinear hysteretic model (Model 3) are rotational springs, and not translational springs. The presentation of Model 3 in Figure 5 will be updated in the revised version.

**7. There is some confusion with respect to the pile length below mudline. Figure 4 tells the reader that it is 10 m, whereas on the next page a value of 9.5 m is mentioned. It is also unclear to the reader why this particular value was chosen and if the results are sensitive to this aspect of the model.**

Thank you for noticing this typo. Model 3 is applied at 10 m below mudline. We will correct the text accordingly.

The moment-rotation curve at mudline is not very sensitive to a change in application point of Model 3, whereas the moment-horizontal displacement curve at mudline is sensitive. If Model 3 is applied 1 m deeper, that is at 11 m instead of at 10 m, then the moment-rotation curve will increase by 2%, while the moment-horizontal displacement curve will increase by 10%.

**8. Section 3.2.3: "Masing's rule" should be explained to readers not familiar with it.**

The following text will be included in Section 3.2.3, around line 7:

"Masing (1926) proposed that, if the force $f$ vs. displacement $x$ curve for system at initial loading is described by the relation $f = f(x)$, then the unloading and reloading branches of the response of the system are geometrically similar to the initial-loading curve, but factored by 2, and are described by

$$\frac{f - f_0}{2} = f\left(\frac{x - x_0}{2}\right)$$

Where $(x_0, f_0)$ is the load reversal point for that particular loading branch."

**9. Section 3.2.4: It is unclear which degrees of freedom are affected by this model. In particular, is it a 1D or 2D model? It is also unclear how the bottom of the pile is treated. Typically this is modelled as being clamped to the ground, to guarantee suitable boundary conditions for finite element analyses, but this seems not to be the case here?**

Thank you for your comment, we have realized that the paper does not state clearly to which degrees of freedom Model 4 (*p-y* curves) is applied. This will be clarified for all the soil-foundation models in the revised version. 1D *p-y* curves are applied uncoupled at the horizontal degrees of freedom (longitudinal and transversal) along the depth of the embedded pile. The bottom of the pile is restrained for vertical displacement, and rotation around the vertical axis.

**10. The order of the models is somewhat unnatural. The classical p-y approach comes last, whereas it should probably be the first model. The new nonlinear model occurs in between, whereas it should probably be the last model.**

Thank you for pointing out this, we agree that this is an unnatural order. The order of the models will be rearranged according to: 1. *P-y* curves, 2.Linear elastic, 3. Linear elastic with damping, 4. Non-linear with hysteresis.

**11. Environmental data were taken from a 25 m site, but the study assumes 20 m of water depth. Why?**

The geometry and characteristics of the offshore wind turbine, the water depth and the soil conditions used in this study were based on Phase II of the OC3 comparison exercise. In the OC3 exercise, a water depth of 20 m was considered. To find realistic environmental conditions, also with a limited need for simulation time, the UpWind lumped scatter diagram (Fischer et al. 2010) was considered a good alternative due to its availability, even though the water depth differs with 25%.

**12. Section 4.4.1: "coupling effects between the two horizontal axes are neglected ... since mainly in plane loads are consider, the simplification is considered to be acceptable" - Does this reasoning also apply to the (important) idling load cases?**

This is a relevant comment. As an example, Figure 6 shows the trajectory of the mudline node for LC1 and LC15. The assumption holds well for LC1, but not for LC15, with transversal motions up to about 2/3 of the longitudinal motions. This OK for a linear stiffness matrix (model 1 and 2) applied separately to the x and y components of the motions, but the nonlinear models will have additional coupling terms. These terms are currently not implemented. LC15 does not contribute much to the total fatigue, but we will nevertheless make an assessment of the impact of coupling for all the load cases in the revised paper.

[Figure]

Figure 6: Trajectory of the mudline node in the horizontal plane for LC1 and LC15.

**13. Figure 10: What are the scales? What are the units?**

Figure 10 is only conceptual and does not give the real moment-rotation curves for the different models - therefore no units and scales. The figure is included to illustrate the differences in stiffness characteristics between the soil-foundation models. We understand that this can be misleading, and will therefore make a new plot with the real moment-rotation curves for the models, including units and scales.

**14. Figures 12-13: There seems to a mistake with the scales and units. The x-axes are similar, but with an order of magnitude difference? The y-axis in Figure 13 should read "Horizontal displacement ..."? The figure is also not displaying a moment-displacement curve as reported in the caption, is it?**

Thank you for noticing this. There are some errors in these figures will be corrected. The revised Figures 12 and 13 of the paper are shown in Figure 7 and 8, respectively.

[Figure]

*Figure 7: Computed moment – rotation curve at mudline from finite element analyses and from the calibrated Model 3. The representative moment and rotation at mudline used in the calibration from Passon (2006) are included as a reference.*

[Figure]

Figure 8: Computed moment – horizontal displacement curve at mudline from finite element analyses and from the calibrated Model 3. The representative moment and rotation at mudline used in the calibration from Passon (2006) are included as a reference.

**15. Section 4.5: "Parameter values used in this thesis ..."? If the results in the paper are based on a master thesis, this needs to be mentioned (e.g. in a footnote) and the thesis should be cited.**

The following will be added in the introduction:

"The paper is a continuation of the master's thesis of the first author Aasen (2016), where all co-authors have been involved."

**16. "Section 4.5: "absolute values should be evaluated accordingly" - I assume that you want to say that "the reported values should be evaluated accordingly?" It is not true that "This will not influence relative values ..." the results will be skewed to a certain degree - you should explain your reasoning here in more detail."**

Thank you for pointing this out. We agree that the results will be skewed when this part of the S-N curves is not considered. The calculations in the revised version will be with the full S-N curve, including the high stress region. The text on page 16 (line 22) will be reformulated as:

"The duration of each load case is 1800 seconds. Results have been extrapolated to find the accumulated fatigue damage per year."

**17. Figure 14: What are the eigenfrequencies of the different wind turbine models? The results of the decay test shown suggest that there are differences between the different models? Then the difference in accumulated damage reported in Figure 16 can probably be**

**explained by these frequency differences already. It is unclear what further effects the differences in foundation modelling have. In particular, it is not true then that "it can be seen how the nonlinear foundation stiffness influence fatigue damage".**

Figure 9 shows the frequency as a function of the cycle number for the free vibration test shown in Figures 14 and 15 of the paper. For the cyclic load level amplitude considered in the free vibration test, the models have different stiffness, and therefore slightly different eigenfrequencies.

Figure 9 also shows that the frequency calculated with Model 3 increases with the cycle number, that is, it increases with decreasing cyclic load amplitude. This is a consequence of the non-linear foundation stiffness. If a free vibration test is run with the cyclic amplitude considered in the calibration (about 120 MNm), then the calculated frequencies for all the models would be similar in the first load cycles before non-linearity enters the picture.

[Figure]

*Figure 9: Frequency vs. cycle number from the free vibration test shown in Figure 14 of the paper*

The difference in eigen frequencies shown in Figure 9 is then a consequence of the chosen approach of model comparison. It was decided to base all models on the stiffness and load levels from Passon (2006). However, we agree to the reviewer's comment that the paragraph: **"it can be seen how the nonlinear foundation stiffness influence fatigue damage"** is too simple, since the effect comes from both nonlinearity in the working load range and different stiffness. However, the difference in stiffness is as well a consequence of the nonlinearity. We will modify the paragraph to:

"It can be seen how the nonlinear foundation model influences fatigue damage. The effect is partly explained by the non-linear behavior (which generates differences in foundation stiffness and damping) in the load range considered and partly by the inherent difference in stiffness at fatigue load level compared to the levels considered by Passon. While a slightly larger stiffness in Models 1 and 2 could have moderated this conclusion, the results support the value of using non-linear hysteretic models that automatically account for the soil response in different load regimes."

As mentioned earlier, the differences in initial stiffness is important for LC1. For the other load cases contributing to fatigue, the tangent stiffness at the higher load levels is more important. The "mean" tangent stiffness during operation is relatively similar for all models.

**18. Figure 17: Comparing the damage for the idling load cases, it is unclear which of the wind turbine models can move in the side-to-side direction and which cannot - it seems that some foundation models restrict the relevant degrees of freedom, for example, whereas others do not. This could already explain a large amount of the differences and should be discussed in more detail. Especially since "this gives a significant contribution to the total fatigue damage" - please report quantitatively how large this contribution is, instead of just saying that it is "significant", which has no objective meaning here.**

Thank you for pointing this out. This emphasizes the need to clarify in which directions the different models can move. All four soil-foundation models are free to move in the fore-aft and side-to-side direction. Still, because wind and waves are applied in the same plane (x-z plane), the movements are predominantly in this plane (see also answer to comment 12).

The word "significantly" will be substituted by the actual percentage of the total fatigue damage.

**19. Figure 15 and Section 5.2: What are Models 2a and 2b? These should be introduced in Section 3.2, where the methods are described, not in the results section.**

Model 2a, 2b and 2c are conceptually the same, and they only differ in the damping magnitude of the rotational dampers. As they are conceptually the same, we found it natural to present the different calibrations of the model in section 4.4 (Calibration of soil-foundation models). We suggest that we refer to section 4.4.2 in section 3.2 for the different calibrations of the model.

**20. Section 5.2: "On these locations rotor dynamics dominate the loading, which are not considerably influenced by the soil-foundation response." - This is somewhat surprising, as I would expect differences in damping (from the soil) and the eigen-frequencies of the structure to influence these results. Why is this not the case?**

The differences in tower top motions due to foundation differences are not large enough to influence the rotor fatigue loads, driven by 3P tower blockage effects, 1P gravity loads and wind shear (for the blade roots) and turbulence. Relatively small changes in overall rotor loads, however, will influence the bending moments and stresses at the mudline, due to the long moment arm. If desired, this can be explained further by PSD plots of stresses at the blade root, yaw bearing and mudline.

**Technical comments**

Thank you for detailed comments on technical matters. Please see below our replies to the technical comments that need feedback:

**p10, l10: One should focus on describing the model used to describe the wind, not on the particular implementation in specific software.**
The turbulence model used for generating the wind will be addressed.

**Table 1: Seems to contain a typo. The first probability of occurrence should be "0.00671"**

There is a typo, the correct value is 0.06071.

**p13, l4: "the foundation stiffness for Model 1 is independent of load level" - this is presumably also the case for Model 2?**

Yes, this is also the case for Model 2.

**Figure 12: Is this the same data as in figure 10?**

Both figures are representing the rotational stiffness for Model 3. Still figure 10 is only conceptual to illustrate the differences in stiffness characteristics between the soil-foundation models. Figure 12 illustrates real data for Model 3.

**References**

Aasen, S. (2016). Soil-structure interaction modelling for an offshore wind turbine with monopile foundation Master's thesis, Norwegian University of Life Sciences.

Achmus, M., Y.-S. Kuo and K. Abdel-Rahman (2009). "Behavior of monopile foundations under cyclic lateral load." Computers and Geotechnics **36**(5): 725-735.

Benz, T. (2007). Small-strain stiffness of soils and its numerical consequences, Univ. Stuttgart, Inst. f. Geotechnik.

Darendeli, M. B. (2001). Development of a new family of normalized modulus reduction and material damping curves.

Fischer, T., W. De Vries and B. Schmidt (2010). UpWind Design Basis (WP4: Offshore foundations and support structures), Upwind.

Lesny, K. and J. Wiemann (2006). Finite-element-modelling of large diameter monopiles for offshore wind energy converters. Geo Congress.

Masing, G. (1926). Eigenspannungen und verfestigung beim messing. Proceedings of the 2nd international congress of applied mechanics, Zürich.

Page, A. M., S. Schafhirt, G. R. Eiksund, K. S. Skau, H. P. Jostad and H. Sturm (2016). Alternative Numerical Pile Foundation Models for Integrated Analyses of Monopile-based Offshore Wind Turbines. The Proceedings of the Twenty-sixth (2016) International Offshore and Polar Engineering Conference, International Society of Offshore & Polar Engineers**: 111-119.

Passon, P. (2006). "Memorandum: derivation and description of the soil-pile-interaction models." IEA-Annex XXIIII Subtask **2**.

Rosquoët, F., L. Thorel and Y. Canepa (2004). Horizontal cyclic loading of piles installed in sand: study of the pile head displacement and maximum bending moment.

Schroeder, F. C., A. S. Merritt, J. D. Sørensen, A. Muir Wood, C. L. Thilsted and D. M. Potts (2015). Predicting monopile behaviour for the Gode Wind offshore wind farm. Frontiers in Offshore Geotechnics III. V. Meyer. OSlo, Norway, CRC Press.

Sheil, B. B. and B. A. McCabe (2016). "Biaxial Loading of Offshore Monopiles: Numerical Modeling." International Journal of Geomechanics: 04016050.

Zaaijer, M. B. (2006). "Foundation modelling to assess dynamic behaviour of offshore wind turbines." Applied Ocean Research **28**(1): 45-57.